# Spatiotemporal analysis of glioma heterogeneity reveals COL1A1 as an actionable target to disrupt tumor progression

Andrea Comba [1,2,3], Syed M. Faisal [1,2,3], Patrick J. Dunn[1,2,3], Anna E. Argento [1,2], Todd C. Hollon[1], Wajd N. Al-Holou[1], Maria Luisa Varela [1,2,3], Daniel B. Zamler [1,2,3], Gunnar L. Quass [4], Pierre F. Apostolides[4,5], Clifford Abel II[1,2,3], Christine E. Brown [6], Phillip E. Kish [1,7], Alon Kahana[7], Celina G. Kleer [3,8], Sebastien Motsch [9], Maria G. Castro [1,2,3] & Pedro R. Lowenstein [1,2,3,10]✉

Intra-tumoral heterogeneity is a hallmark of glioblastoma that challenges treatment efficacy. However, the mechanisms that set up tumor heterogeneity and tumor cell migration remain poorly understood. Herein, we present a comprehensive spatiotemporal study that aligns distinctive intra-tumoral histopathological structures, oncostreams, with dynamic properties and a specific, actionable, spatial transcriptomic signature. Oncostreams are dynamic multicellular fascicles of spindle-like and aligned cells with mesenchymal properties, detected using ex vivo explants and in vivo intravital imaging. Their density correlates with tumor aggressiveness in genetically engineered mouse glioma models, and high grade human gliomas. Oncostreams facilitate the intra-tumoral distribution of tumoral and non-tumoral cells, and potentially the collective invasion of the normal brain. These fascicles are defined by a specific molecular signature that regulates their organization and function. Oncostreams structure and function depend on overexpression of COL1A1. *Col1a1* is a central gene in the dynamic organization of glioma mesenchymal transformation, and a powerful regulator of glioma malignant behavior. Inhibition of *Col1a1* eliminates oncostreams, reprograms the malignant histopathological phenotype, reduces expression of the mesenchymal associated genes, induces changes in the tumor microenvironment and prolongs animal survival. Oncostreams represent a pathological marker of potential value for diagnosis, prognosis, and treatment.

[1] Department of Neurosurgery, University of Michigan Medical School, Ann Arbor, MI 48109, USA. [2] Department of Cell and Developmental Biology, University of Michigan Medical School, Ann Arbor 48109 MI, USA. [3] Rogel Cancer Center, University of Michigan Medical School, Ann Arbor, MI 48109, USA. [4] Kresge Hearing Research Institute, Department of Otolaryngology-Head & Neck Surgery, University of Michigan Medical School, Ann Arbor, MI 48109, USA. [5] Department of Molecular & Integrative Physiology, University of Michigan Medical School, Ann Arbor, MI 48109, USA. [6] Department of Hematology & Hematopoietic Cell Transplantation, National Medical Center, City of Hope, Duarte, CA 91010, USA. [7] Ophthalmology & Visual Science, University of Michigan Medical School, Ann Arbor, MI 48109, USA. [8] Department of Pathology, University of Michigan Medical School, Ann Arbor, MI 48109, USA. [9] School of Mathematical and Statistical Sciences, Arizona State University, Tempe, AZ 85287, USA. [10] Department of Biomedical Engineering, University of Michigan, Ann Arbor, MI 48109, USA. ✉email: pedrol@umich.edu

High grade gliomas (HGG) are the most prevalent and malignant brain tumors. They grow rapidly, invade surrounding normal brain, and recur within 12 months. Median survival is 18–20 months, in spite of current standard of care[1,2]. Despite some notable successful outcomes from the large cancer sequencing program, which identified driver genes in a number of cancers, effective therapeutically actionable breakthroughs have not yet been identified in HGG[3–7].

HGG are highly heterogeneous at the histological, cellular, and molecular level. Heterogeneity of HGG is illustrated in addition by characteristic pathological structures such as pseudopalisades, microvascular proliferation, and areas of hypoxia and necrosis[2,8]. The molecular characterization of glioma heterogeneity identified three main molecular signatures: proneural, mesenchymal, and classical[4,9]. However, later studies demonstrated that all three transcriptomic signatures are expressed within individual tumors[5,10,11]. Rather than outright glioma subtypes, the consensus proposes that individual tumors are enriched in particular molecular subtypes. Thus, studies have correlated histological features with genetic alterations and transcriptional expression patterns. For example, highly aggressive histological features such as hypoxic, necrotic and microvascular proliferative zones have been associated with the mesenchymal molecular signature and worse prognosis[8]. However, the molecular classification has only minor clinical impact. Thus, alternative classification schemes using a pathway-based classification are currently being considered[12]. How these new classifications will deal with tumor heterogeneity remains to be explored. Moreover, different microenvironmental, metabolic, and therapeutic factors drive transitions of the GBM transcriptomic signature, particularly transitions to mesenchymal states. It is important to note that glioblastoma plasticity explains the high degree of tumor heterogeneity and favor the selection of new clones at recurrence or therapy resistance[13–16]. It has been proposed that intra-tumoral heterogeneity is represented by four main cellular states, the progenitor, astrocyte, oligodendrocyte, and mesenchymal like-state, which are a consequence of tumor plasticity and are affected by the tumor microenvironment[15].

Tumor mesenchymal transformation is a hallmark of gliomas[13,17,18]. A mesenchymal phenotype is defined by cells with spindle-like, fibroblast-like morphology associated with alterations in their dynamic cellular organization leading to an increase in cell migration and invasion[19,20]. The mesenchymal phenotype is controlled by particular transcription factors and downstream genes related to the extracellular matrix (ECM), cell adhesion, migration, and tumor angiogenesis[18,21,22].

However, the cellular and molecular mechanisms that regulate mesenchymal transformation in gliomas, especially concerning the mesenchymal features of invasive cells, has remained elusive. Integrating morphological features, spatially resolved transcriptomics, and cellular dynamics resulting from mesenchymal transformation, growth, and invasion are thus of great relevance to the understanding of glioma progression[13,20].

Cell migration is essential to continued cancer growth and invasion. Morphological and biochemical changes that occur during mesenchymal transformation allow glioma cells to move throughout the tumor microenvironment and invade the adjacent normal brain. Tumor cells also migrate along blood vessels, white matter tracks, and the subpial surface[23]. Within the tumor microenvironment, aligned extracellular matrix fibers help guide the movement of highly motile mesenchymal-like breast cancer cells[24,25].

Collective motion is a form of collective behavior where individual units' (cells) movement is regulated by local intercellular interactions (i.e., attraction/repulsion) resulting in large scale coordinated cellular migration[26–30]. Collective motion plays an essential role in embryogenesis and wound healing[27,31–33]. Emergent organized collective motion patterns could help explain so far poorly understood tumoral behaviors such as invasion, metastasis, and especially recurrence[27,31].

Studies of tumor motility have concentrated on the behavior of glioma cells at the tumor invasive border[27,34,35]. Potential motility at the glioma core has not been studied in much detail so far. This study challenges the conventional belief that cells in the central core are non-motile and indicate that the glioma core displays collective migratory patterns. This would suggest that the capacity of gliomas to invade and grow, results from phenomena occurring at the tumor invasive border, and from the overall capacity of gliomas to organize collective motion throughout the tumor mass, from the tumor core to the tumor invasive border with normal brain.

Collagen1a1 (COL1A1) has been shown previously to be a major component of the extracellular matrix in different cancers, including glioma, and has been reported to promote tumor growth and invasion[36,37]. Alternatively, some data suggest that collagen fibers could be passive barriers to resist tumor cell infiltration, or provide biophysical and biochemical support for cell migration. Some studies reported that density of COL1A1 inversely correlates with glioma patients' prognosis[38]. However, other studies showed that either increased or decreased deposition of collagen could be associated with increased tumor malignancy[39–41]. Therefore, it is important to further determine the role of COL1A1 in glioma growth and invasion.

Our study reveals that malignant gliomas, both high grade human gliomas and mouse glioma models, display regular distinctive anatomical multicellular fascicles of aligned and elongated, spindle-like cell. We suggest they are areas of mesenchymal transformation. For the sake of simplicity in their description throughout the manuscript, we have named these areas 'oncostreams'. Using time-lapse laser scanning confocal imaging of high grade glioma explants ex vivo, and multiphoton microscopy in vivo we demonstrate that oncostreams are organized collective dynamic structures. They are present at the tumor core and at areas of the tumor border at the interphase with the normal brain. To study the molecular mechanisms underlying oncostream organization and function we use laser capture microdissection (LCM) followed by RNA-sequencing and bioinformatics analysis. We discovered that oncostreams are defined by a mesenchymal transformation signature enriched in extracellular matrix-related proteins, and which suggests that COL1A1 is a key determinant of oncostream organization. Inhibition of *Col1a1* within glioma cells led to oncostream loss and reshaping of the highly aggressive phenotype of HGG. These data indicate that COL1A1 contributes to the tumor microenvironment scaffold, and serves to organize areas of collective motion in gliomas. In summary, we provide a comprehensive study of the histological, morphological, and dynamic properties of glioma tumors. In addition, we characterize the molecular mechanisms that define intra-tumoral mesenchymal transformation in gliomas and discuss their therapeutic implications. Oncostreams are anatomically and molecularly distinctive, regulate glioma growth, display collective motion, and are regulated by the extracellular matrix, especially by COL1A1. Inhibiting *Col1a1* within glioma cells is a potential therapeutic strategy to mitigate glioma mesenchymal transformation, intra-tumoral heterogeneity, and thus, potentially reduce deadly glioma invasion and continued growth.

## Results

**Intra-tumoral multicellular fascicles of elongated and aligned cells in gliomas: oncostreams**. HGG are characterized by anatomical, cellular, and molecular heterogeneity which determines,

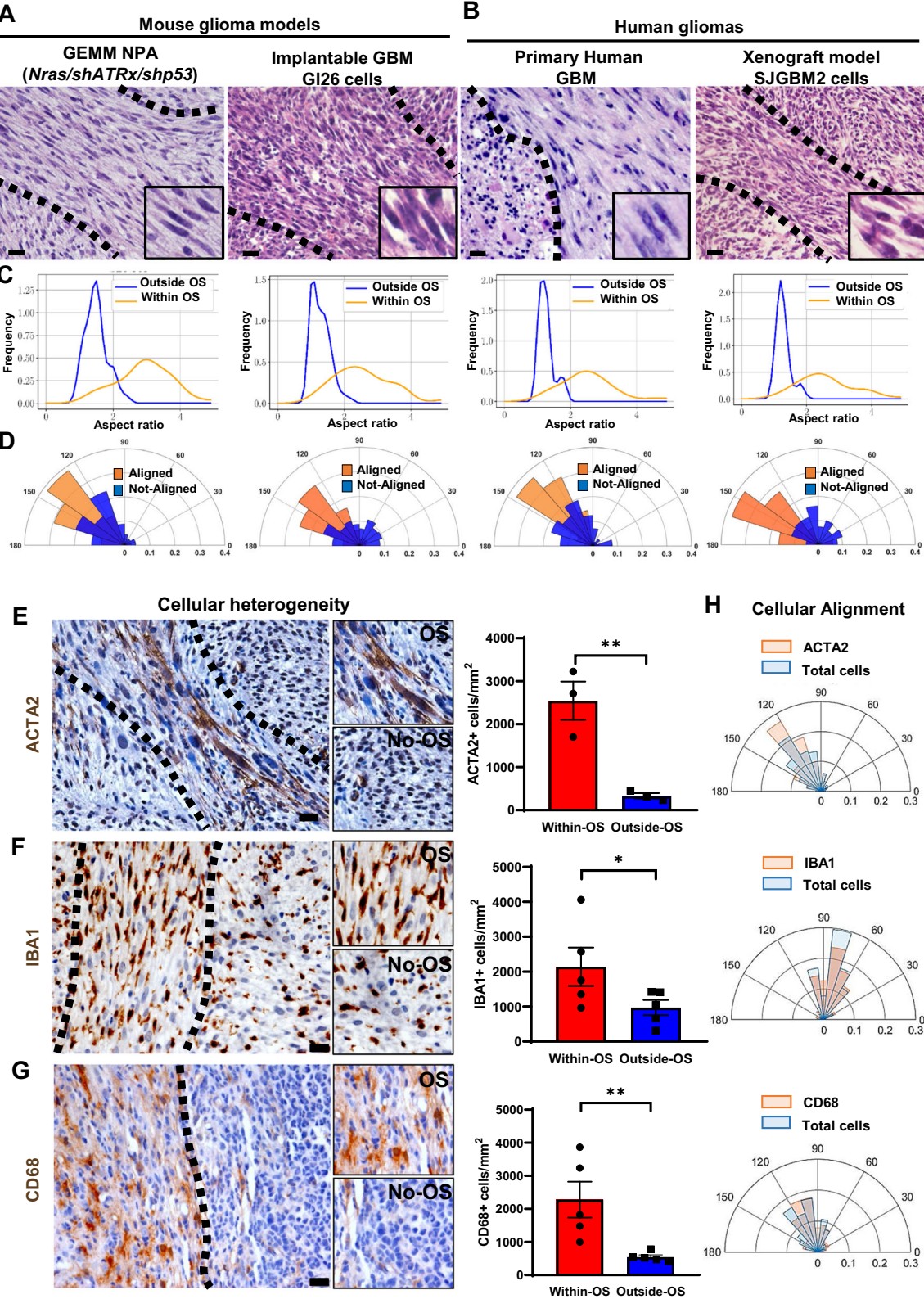

in part, tumor aggressiveness and reduces treatment efficacy[5,7,11]. Histopathological analysis of mouse and human gliomas revealed the presence of frequent distinct multicellular fascicles of elongated (spindle-like) and aligned cells (≈5–30 cells wide) distributed throughout the tumors. These structures resemble areas of mesenchymal transformation which we describe as oncostreams (Fig. 1A, B).

To study the presence and morphological characterization of oncostreams, we examined histological sections from various mouse glioma models as well as human glioma specimens (Fig. 1A, B). We determined the existence of oncostreams in genetically engineered mouse models (GEMM) of glioma including NPA (*NRAS, shP53, shAtrx*) and NPD (*NRAS, shP53, Pdgfb*) and other implantable models (GL26) (Fig. 1A and

**Fig. 1 Oncostreams are multicellular fascicles present in mouse and human gliomas. A** Representative 5 μm Hematoxylin and Eosin (H&E) microtome sections from gliomas showing that fascicles of spindle-like glioma cells (oncostreams, outlined by the dotted line) are present in a Genetically Engineer Mouse Models of gliomas NPA (*NRAS*/sh*Atrx*/sh*P53*) and the GL26 intracranial implantable model of glioma. Sections from more than 10 tumors per group were analyzed. Scale bars: 50 μm. **B** Representative H&E microtome sections of human glioma and human xenografts showing the presence of oncostreams. Sections from more than 5 tumors per group were analyzed. Scale bar: 20 μm. Histograms showing the cellular shape analysis (aspect ratio) (**C**) and angle orientation (alignment) for the corresponding images (**D**) show areas of oncostreams (OS) formed by elongated and aligned cells and areas with no oncostreams (No-OS) as rounded and not-aligned cells. **E, G** Immunostaining shows that tumor cells, mesenchymal cells (ACTA2+), microglia/macrophages (IBA1+ and CD68+), are aligned within, the main orientation axis of oncostreams. Bar graphs show the quantification of ACTA2 + cells, *n* = 3 (\*\**p* = 0.0055) (**E**), IBA1+ cells, *n* = 5 (\**p* = 0.0252) (**F**) and CD68+ cells, *n* = 5 (\**p* = 0.051) (**G**) within oncostreams areas in NPA tumors. 6–13 areas of oncostreams per tumor section per animal were imaged. Graphs present mean ± SEM; Paired two-sided t-test analysis. Scale bar: 20 μm. **H** Angle orientation shows the alignment of ACTA2+, IBA1+, and CD68+ cells within oncostreams for the corresponding images. Source data are provided as Source data.

Supplementary Fig. 1A, B). Moreover, human glioma samples from primary resections and a xenograft glioma model, SJGBM2, established the presence of these multicellular structures in human tissue (Fig. 1B and Supplementary Fig. 1C). Morphological analysis determined that cells within histological areas corresponding to oncostreams have an aspect ratio of 2.63 ± 0.19 (elongated or spindle-like cells) compared to the surrounding tissue where cells have an aspect ratio of 1.37 ± 0.12 (round cells), both in mouse and human gliomas as shown in Fig. 1C and Supplementary Fig. 1D. We also determined that elongated cells within oncostreams are nematically aligned with each other, whereas outside of oncostreams, cell orientations are not aligned (Fig. 1D and Supplementary Fig. 1E).

To gain insight into the cellular features of oncostreams we asked if they are homogeneous or heterogeneous multicellular structures. We observed that in GEMM of gliomas, oncostreams are formed by GFP+ tumor cells, and are enriched in other tumor microenvironment cells such as ACTA2+ mesenchymal cells, IBA1+, and CD68+ tumor associated microglia/macrophages cells, Nestin+ cells and GFAP+ glial derived cells (Fig. 1E–G and Supplementary Fig. 2A–D). The quantification of mesenchymal cells (ACTA2+), and tumor associated microglia/macrophages (TAM) cells (CD68+ and IBA1+) showed a significant enrichment of these populations within oncostreams compared to the surrounding areas (Fig. 1E–G). Moreover, non-tumoral cells within oncostreams were positively aligned along the main axes of oncostreams, and with tumor cells in mouse gliomas (Fig. 1H). This suggests that oncostreams are mesenchymal-like structures which interact with TAM and mesenchymal cells.

To test if oncostreams form along existing brain structures, we evaluated their co-localization with white matter tracts. Although, occasional positive immune-reactivity (Neurofilament-L) was present within some areas of the tumors, oncostream fascicles were not preferentially organized along brain axonal pathways (Supplementary Fig. 1F). These data indicate that oncostreams are fascicles of spindle-like aligned cells within glioma tumors, which contain tumor and non-tumor cells.

**Oncostream density positively correlates with tumor aggressiveness and poor prognosis in mouse and human gliomas.** Oncostreams are histological features that contribute to intra-tumoral heterogeneity suggesting a potential role in glioma progression and malignancy. To understand whether the presence of oncostreams correlates with tumor aggressiveness and clinical outcomes, we generated genetically engineered tumors of different malignant behaviors using the Sleeping Beauty Transposon system. These models reproduce the malignant histopathological features of gliomas as demonstrated in previous studies[42–45]. We induced tumors harboring two different genotypes: (1) Activation of *Rtk/Ras/Pi3k* pathway, in combination with *P53* and *Atrx*

downregulation (NPA), and, (2) *Rtk/Ras/Pi3k* activation, *P53* downregulation, *Atrx* downregulation, and mutant *IDH1-R132* expression (**NPAI**) (Fig. 2A). *IDH1*-wild-type tumors (NPA) display a highly malignant phenotype and worse survival prognosis (Mediam survival (MS): 70 days), compared with tumors harboring the *IDH1-R132* mutation, **NPAI**, (MS: 213 days) (Fig. 2B). This outcome reproduces human disease, as patients with IDH1-Mutant tumors also have prolonged median survival[1,45,46]. Tumor histopathological analysis showed a positive correlation between the density of oncostreams and tumor malignancy (Fig. 2C, D). NPA (IDH1-WT) tumors exhibited larger areas of oncostreams within a highly infiltrative and heterogeneous glioma characterized by abundant necrosis, microvascular proliferation, pseudopalisades and cellular heterogeneity as described before[44,45]. Conversely, NPAI (*IDH1-Mut*) tumors display a very low density of oncostreams and a homogenous histology mainly comprised of round cells, low amounts of necrosis, no microvascular proliferation, absence of pseudopalisades and less invasive borders (Fig. 2C and Supplementary Fig. 5).

Further, to objectively identify and quantify tumor areas covered by oncostreams, we trained a fully convolutional neural network (fCNN) (Supplementary Figs. 3 and 4A). Our deep learning analysis found that oncostreams occupied 15.28 ± 6.10% of the area in NPA tumors compared with only 1.18 ± 0.81% in NPAI tumors (Fig. 2C, D and Supplementary Fig. 5A, B). Cellular alignment analysis validated the presence or absence of oncostreams (Fig. 2E).

To determine whether oncostreams are linked to glioma aggressiveness in human patients, we evaluated a large cohort of TCGA glioma diagnostic tissue slides from the Genomic Data Commons Portal of the National Cancer Institute. We visually examined 100 TCGA-glioblastoma multiforme (GBM) tissue sections (WHO Grade IV) and 120 TCGA-low grade glioma (LGG) tissues (WHO Grade II and III) using the portal's slide image viewer (Supplementary Table 1). Oncostreams were present in 47% of TCGA-GBM grade IV tumors tissue, in 8.6% of TCGA-LGG grade III, and were absent from TCGA-LGG grade II (Fig. 3A–C and Supplementary Data 1), consistent with tumor aggressiveness (http://gliovis.bioinfo.cnio.es)[47]. We then determined the presence of oncostreams across known molecular subtypes of HGG (Grade IV)[4]. We found oncostream fascicles in 59.4% of Mesenchymal (MES), 53.6% of Classical (CL) subtypes, and only 26.7% of Proneural (PN) (Supplementary Fig. 6A). Finally, we evaluated oncostreams presence related to IDH status and 1p 19q co-deletion in LGG (Grade III). Oncostreams were present in 16.6% of IDH-WT subtype, 5% of IDHmut-non-codel, and absent from IDHmut-codel subtype (Supplementary Fig. 6B). These analyses suggest that oncostream presence is higher in Mesenchymal and Classical subtypes and correlates with IDH-WT status, and thus with a poor prognosis.

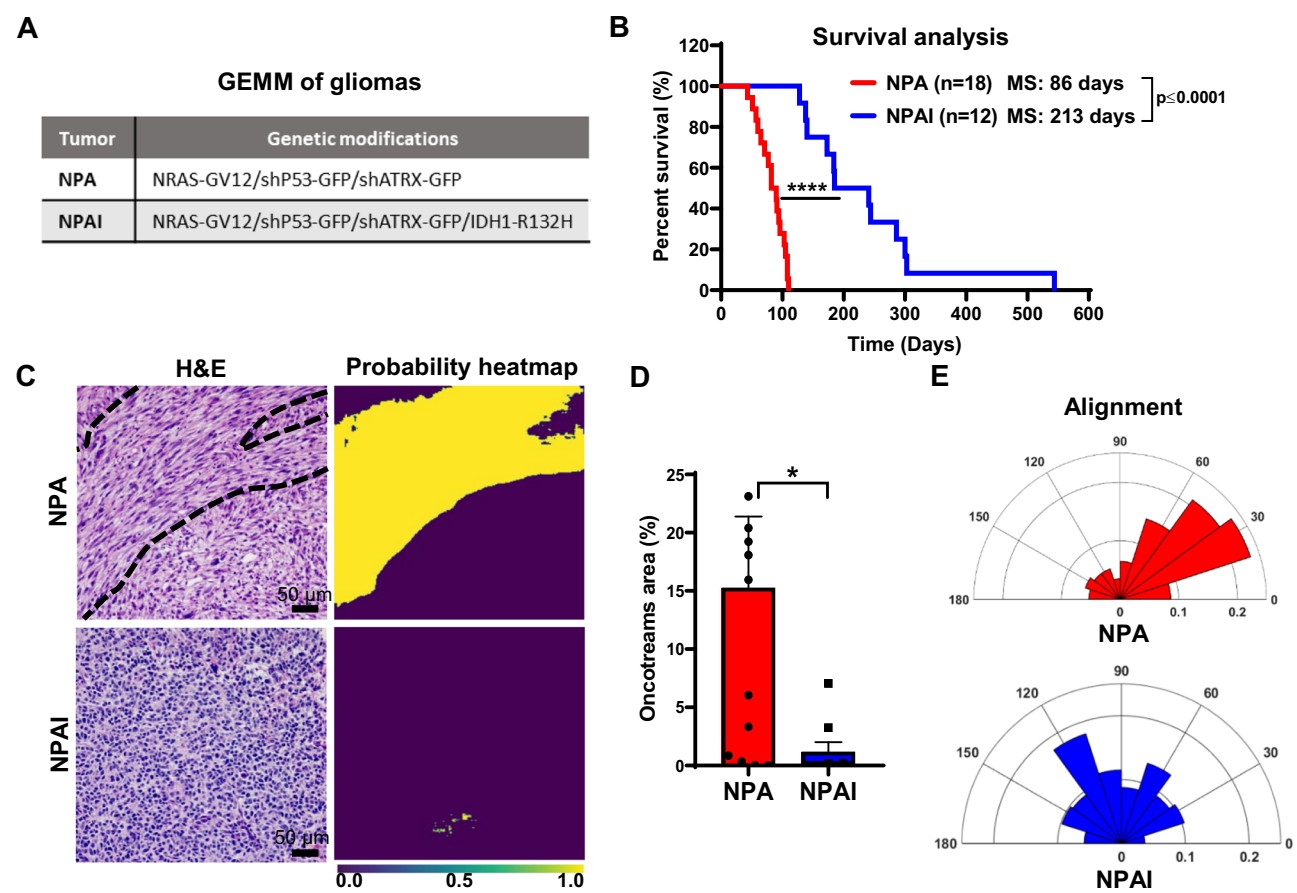

**Fig. 2 Oncostreams density positively correlates with tumor aggressiveness in GEMM of gliomas. A** Genetic makeup of NPA and NPAI tumors. **B** Kaplan–Meier survival curves of NPA and NPAI mouse gliomas show that animals bearing IDH1-R132H mutant tumors (NPAI) have prolonged median survival (MS): **NPA** (MS: 86 days; n:18) versus **NPAI** (MS: 213 days; n:12). Log-rank (Mantel–Cox) test; ****p < 0.0001. **C**, **D** Deep learning method for oncostream detection in Hematoxylin and Eosin (H&E) stained mouse glioma sections: **C** Representative images of oncostreams manually segmented on H&E stained sections. The output of our trained model for each image is shown below (probability heat maps), for tissues containing oncostreams (NPA), and without oncostreams (NPAI), scale bar :50 μm. Quantification of (**C**) is shown in (**D**). **D** 10–14 random fields per tumor section per animal were imaged; independent biological samples n = 12 NPA and n = 9 NPAI were quantified using deep learning analysis. Graphs present mean ± SEM; Mann–Whitney two-sided t-test, *p = 0.0104. **E** Angle histogram plots show aligned cells in NPA tumors vs non-aligned cells in NPAI tumors for the representative images showed in figure. Source data are provided as Source data.

To validate the histological identification, we examined H&E images using our deep learning algorithm (Supplementary Fig. 4B). We observed a strong concordance (>80%) between machine learning and the manual histological identification of oncostreams (Supplementary Table 2). Oncostream presence and their segmentation by deep learning is illustrated in Fig. 3C and Supplementary Figs. 7 and 8. Additionally, alignment analysis of glioma cells confirmed the existence of fascicles of elongated, mesenchymal-like cells in human gliomas (Fig. 3D). Thus, our deep learning algorithm validates our histological identification of oncostreams and confirms that the density of oncostream fascicles positively correlates with glioma aggressiveness.

The analysis of cellular heterogeneity showed that non-tumoral cells such as IBA1+ macrophages/microglia and GFAP+ glial-derived cells were positively aligned within oncostreams tumoral cells (SOX2+) in human HGG (Fig. 3E). Conversely, we detected that low grade gliomas (LGG) exhibited homogenous round cells, GFAP+, and IBA1+ cells throughout the tumor with no defined orientation or alignment (Fig. 3F).

**Oncostreams are defined by a distinctive spatial transcriptome signature.** To determine whether oncostreams fascicles are characterized by a specific gene expression profile, we performed

a spatially-resolved transcriptomic analysis using LCM coupled to RNA sequencing (RNA-Seq). Oncostreams were dissected according to their morphological characteristics defined above. Surrounding areas of homogenous rounded cells were selected as non-oncostreams areas (control) (Fig. 4A). RNA-Seq analysis detected a set of 43 differentially expressed (DE) genes; 16 genes were upregulated and 27 downregulated within oncostreams (Fig. 4B, C and Supplementary Data 2).

Functional enrichment analysis of DE genes, performed using the I-PathwayGuide platform (Advaita Corporation, MI, USA), showed that False Discovery Rate (FDR) corrected gene ontology (GOs) terms were associated with migration and extracellular matrix biological process. GOs such as "positive regulation of motility", "positive regulation of cell migration", "collagen catabolic processes" and "extracellular matrix organization" were the most over-represented biological processes (Fig. 4D and Supplementary Table 3). The upregulated DE genes within the relevant GOs include: *Col1a1, Mmp9, Mmp10, Acta2, Adamts2, Cdh5, Cyr61, Plp1* and those downregulated were *Enpp2, Akap12, Bdkrb1* (Fig. 4E and Supplementary Fig. 9). Significant DE genes shared by related GOs are shown in Supplementary Fig. 9. These data indicate that oncostreams can be identified by a specific gene expression set and suggest a distinct role for oncostreams as

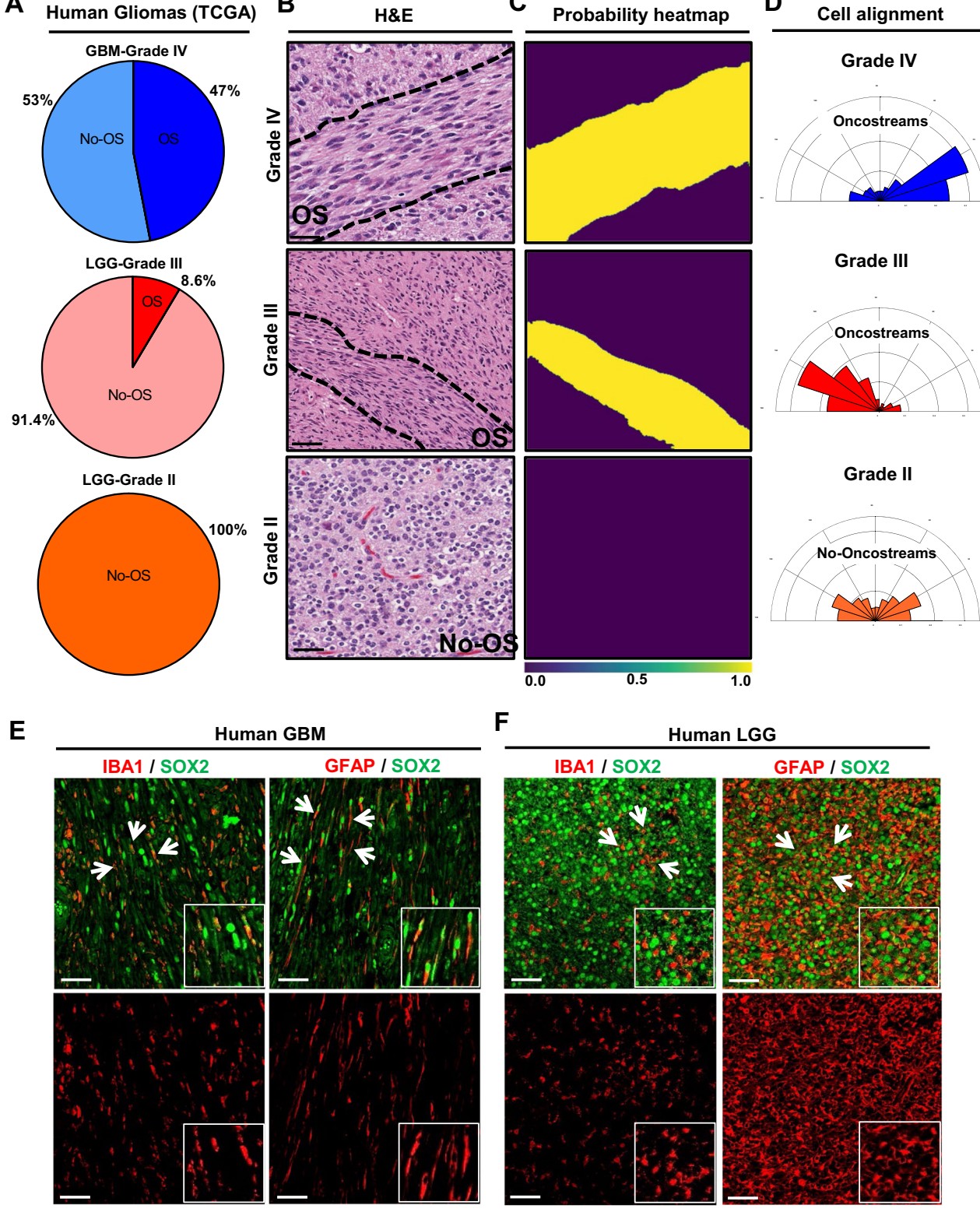

intra-tumoral mesenchymal-like migratory assemblies within glioma tumors.

**COL1A1 contributes to oncostream organization in high grade gliomas.** Histopathologically, oncostreams are spindle-like multicellular fascicles with a defined DE gene expression signature enriched in mesenchymal genes. The GO ontology analyses suggest a central role of collagen catabolic process and extracellular matrix organization in oncostreams function. To understand the molecular mechanisms that regulate oncostream organization and function, we identified critical genes using network analysis. Network interactions revealed that *Col1a1* is a hub gene, one of the most highly connected nodes, representing a potential regulator of the network's signaling pathways and biological functions (Fig. 5A and Supplementary Fig. 10A). We found that the most relevant *Col1a1*-related pathways include:

**Fig. 3 The density of oncostreams positively correlates with tumor aggressiveness in human gliomas. A** TCGA tumors were analyzed from different grade: GBM-Grade IV (100 tumors). LGG-Grade III (70 tumors) and LGG-Grade II (50 tumors). Pie charts show percentage of tumors displaying oncostreams in relation to tumor grade. Oncostreams are present in 47% of GBM grade IV tumors, 8.6 % of LGG grade III, and are absent from LGG grade II. **B** Manual identification of oncostreams in Hematoxylin and Eosin (H&E) images are shown for human gliomas with WHO grades IV ($n = 109$), III ($n = 126$), II ($n = 61$) from TCGA. **C** Deep learning analysis for human gliomas. Our algorithm was able to detect oncostreams in grade IV and III gliomas but not in grade II gliomas. **D** Angle histogram plots show the alignment of cells in H&E histology sections of Grade IV and Grade III gliomas' oncostreams and random alignment in grade II glioma sections lacking oncostreams. Angle histograms correspond to the representative images. **E, F** Immuno-fluorescence staining of SOX2+ tumor cells (green), glial fibrillary acidic protein (GFAP+) cells (red), and microglia/macrophage (IBA1+) cells (red) in high grade human glioblastoma (GBM) WHO Grade IV, IDH-WT ($n = 3$) (**E**) and in low grade glioma (LGG), WHO Grade III, IDH-mutant ($n = 3$) (**F**), showing oncostreams heterogeneity and cellular alignment of these cells in human high grade gliomas but not in low grade gliomas (arrows). Scale bars: 50 μm.

Focal Adhesion, Extracellular Matrix Organization, and Integrin Signaling pathways (Supplementary Fig. 10B, C and Supplementary Tables 4 and 5).

To analyze the role of *Col1a1* in oncostream organization, we analyzed COL1A1 expression by immunofluorescence analysis. The *Col1a1* gene encodes for the alpha-1 chain of type I collagen fibers. We observed that collagen fibers were aligned within oncostreams and overexpressed in more aggressive NPA (IDH1-WT) gliomas compared with NPAI (IDH1-Mut) tumors. COL1A1 expression was significantly lower and only found surrounding blood vessels in NPAI (IDH1-Mut) tumors (Fig. 5B, C). Correspondingly, human GBM glioma tumors (IDH1-WT) with high oncostream densities showed prominent alignment of collagen fibers along these fascicles and higher COL1A1 expression compared to LGG (IDH1-Mut) (Fig. 5D, E).

Moreover, TCGA-glioma data indicate that *COL1A1* has differentially higher expression in GBM histological Grade IV. LGG IDH-WT tumors display higher expression of *COL1A1* than IDH1-Mutant. Within the GBM molecular subtype classification[4,9], the Mesenchymal group shows higher expression of *COL1A1* than the Proneural and Classical groups (Supplementary Fig. 11A); the *COL1A1* gene is clearly associated with the mesenchymal subtype. Analysis of patient survival related to *COL1A1* expression showed that mesenchymal GBM subtype displayed a significantly shorter survival (MS: 10.4 months) for *COL1A1* high tumors, compared to *COL1A1* low (MS: 17.9 months) tumors. Classical and Proneural subtypes did not show survival differences associated to *COL1A1* expression (Supplementary Fig. 11B). Thus, oncostreams represent intratumoral mesenchymal-like structures organized along collagen fibers.

**Col1a1 depletion leads to oncostream loss, tumor microenvironment (TME) remodeling and increases in median survival.** To evaluate the functional role of COL1A1 in oncostream formation we generated a *Col1a1*-deficient genetically engineered mouse glioma model. We generated *Col1a1* wild-type, and *Col1a1* knock-down tumors with different genetic backgrounds (Supplementary Fig. 12A–C). *Col1a1* downregulation increased median survival (MS) (Fig. 5F, G). The knockdown of *Col1a1* in NPA tumors (NPAshCOL1A1) increased survival to MS: 123 days, compared to NPA control tumors (MS: 68 days) (Fig. 5F). Similarly, *Col1a1* knockdown in NPD tumors harboring PDGFβ ligand upregulation (NPDshCOL1A1), also exhibited an increased median survival (MS: 98 days) compared to the NPD controls (MS: 74 days) (Fig. 5G).

To further analyze the effects of *Col1a1* downregulation, we evaluated the histopathological features of glioma tumors, quantified oncostream density using deep learning analysis, and evaluated COL1A1 expression within glioma tissues (Fig. 5H, I). We observed that NPA tumors with *Col1a1* downregulation showed a significant reduction of COL1A1 immunoreactivity within tumors; it was only maintained in small areas surrounding

blood vessels (Fig. 5J, K). *Col1a1* inhibition led to oncostream loss and reprogramming of the histopathological tumoral characteristics as evidenced by homogenous round cell morphology, resembling low grade tumors (Fig. 5J, K). Downregulation of *Col1a1* in NPD tumors appeared less effective, with large areas of remaining COL1A1 (Fig. 5H, I). Nonetheless, *Col1a1* was downregulated within tumor cells and oncostream dismantling was significant compared to NPD control. Some oncostream areas remained associated with blood vessels which displayed significant amounts of COL1A1 (Fig. 5J, K).

We analyzed the effect of *Col1a1* depletion on the intrinsic properties of tumoral cells. In vitro studies showed that *Col1a1*-knockdown cells exhibited a significantly decreased cell proliferation and cell migration compared to controls (Supplementary Fig. 13A–D). Also, we observed that intracranial implantation of *Col1a1*-knockdown cells resulted in decreased tumor growth and progression when compared to controls (Supplementary Fig. 13E). In vivo, genetically engineered *Col1a1* knockdown tumors displayed decreased cell proliferation (PCNA+ cells) (Fig. 6A, B), increased apoptosis via activation of Cleaved-Caspase 3, and downregulation of the anti-apoptotic protein Survivin (Supplementary Fig. 14A–C).

Furthermore, to determine whether *Col1a1* downregulation within glioma cells modifies the glioma TME we analyzed changes in tumor-associated macrophages (TAM), endothelial cells, and mesenchymal cells. We found that *Col1a1* knockdown tumors exhibited a decreased recruitment of CD68+ TAM (Fig. 6C, D), impaired CD31+ endothelial vascular proliferation (Fig. 6E, F), and diminished ACTA2+ perivascular mesenchymal cells (Fig. 6G, H). Moreover, inhibition of *Col1a1* within glioma cells led to downregulation of fibronectin expression, a mesenchymal associated extracellular matrix protein (Supplementary Fig. 14E, F) that is associated with a more aggressive phenotype.

In these preclinical animal models, the expression of *Col1a1* was knocked down from the earliest stages of tumor development. Further, to evaluate the effects of the pharmacological degradation of deposited collagen fibers in highly malignant tumors we analyzed explants of brain tumor sections treated with collagenase. We observed that collagenase treatment decreased reticular fibers (general collagen staining), reduced COL1A1 expression and disassemble fibers' alignment along tumoral cells, and caused oncostreams depletion in a dose-dependent manner (Supplementary Fig. 15A–D). These data indicate that oncostream organization and functions are regulated by COL1A1. *Col1a1* knockdown within glioma cells decreased oncostream formation, reprogramed glioma mesenchymal transformation, and remodeled the glioma TME, thus increasing animal survival. *Col1a1* inhibition represents an approach for future translational development.

**Oncostreams' mesenchymal patterns reveal intra-tumoral collective motion in gliomas.** GO analysis indicates that biological processes such as positive regulation of motility/migration are

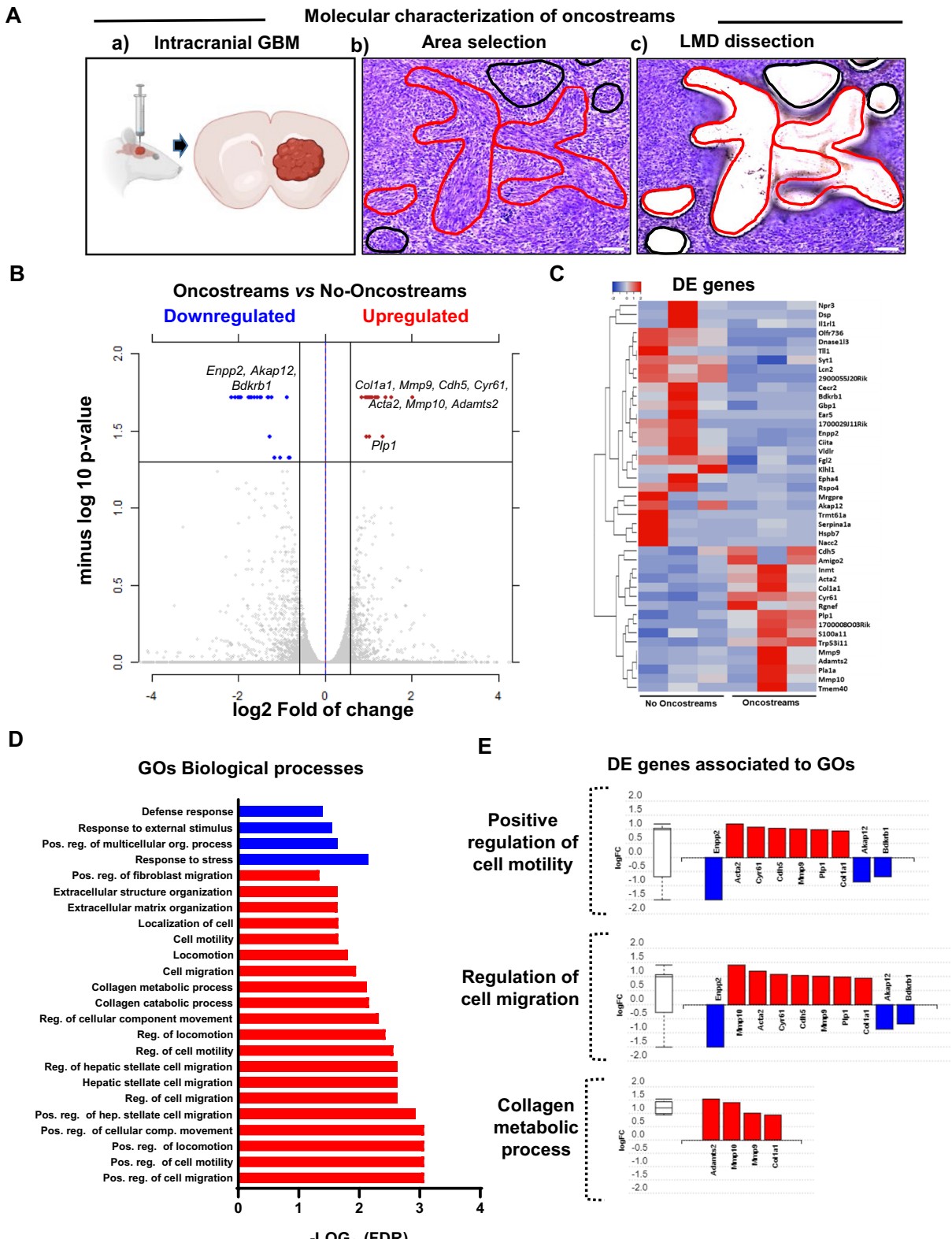

enriched within oncostream fascicles. Overexpression of extra-cellular matrix (ECM)-associated proteins suggest a potential role of COL1A1 fibers in regulating oncostreams' motility. To study if oncostreams represent migratory structures within glioma tumors, we established a physiologically viable explant brain tumor slice model containing a high density of oncostreams (Fig. 7A). The movement of glioma cells expressing green

fluorescent protein (GFP), within the thickness of each explant, was visualized using time-lapse confocal imagining and tracked using Fiji's plug-in Track-Mate (Fig. 7A–C).

Migration analyses show complex glioma cell dynamics throughout the tumor core. The glioma tumor core displays groups of cells (within particular zones) with similar nematic orientation and displaying complex movement patterns (Fig. 7D

**Fig. 4 Oncostreams are defined by a specific gene expression signature related to mesenchymal transformation and migration. A** (a) Schematic representation of spatial transcriptomic analysis of glioma oncostreams using Laser Capture Microdissection (LCM). Glioma tumors were generated by intracranial implantation of NPA tumor cells in C57BL6 mice (Created in BioRender.com). (b, c) Oncostream areas (red outline) were identified and dissected from surrounding glioma tissue (black outline) of three independent mouse tumor samples using a LCM microscope. Scale bar: 100 μm. **B** A volcano plot displays differentially expressed (DE) genes from oncostream *vs* no-oncostream areas. DE genes were selected based on a fold change of ≥ 1.5 and a *q*-value (false discovery rate (FDR) corrected *p*-value) of ≤ 0.05. Two-sided multiple t-test. Upregulated genes (red dots) and downregulated genes (blue dots) are shown. Relevant genes related to mesenchymal migration are labeled on the graph (*n* = 3, biologically independent replicates per group). **C** Heat map illustrates gene expression patterns for oncostream vs no-oncostream areas in NPA glioma tumors (*n* = 3, biologically independent replicates per group). Differentially upregulated genes (16) are represented in red and downregulated genes (*n* = 27) are represented in blue (*q*-value ≤ 0.05 and fold change ≥ ± 1.5). **D** Functional enrichment analysis of overrepresented Gene Ontologies (GO) terms (biological processes) obtained when comparing oncostream vs no-oncostream DE genes. *p*-value corrected for multiple comparisons using the FDR method. Cutoff FDR < 0.05. Blue: Downregulated GOs. Red: Upregulated GOs. **E** Bar graphs show DE genes annotated to the most relevant enriched GOs biological process. Upregulated genes (red), downregulated genes are (blue). Biologically independent replicates (*n* = 3) per group. Box and whisker plot on the left summarizes the distribution of differentially expressed genes. center line of the boxplot (median), upper bound of the boxplot is the Q3 (75th percentile), the lower bound of the boxplot is the Q1(25th percentile), the upper whiskers is the maximum value of the data (within 1.5 times the interquartile range over the 75th percentile), and the lower whisker is the minimum value of the data (within 1.5 times the interquartile range under the 25th percentile). Outliers are represented by circles. The Data (**D**, **E**) were analyzed using Advaita Bio's iPathwayGuide.

and Supplementary Fig. 16A) and, which represent collective motion[27,29–31]. Angle velocity distribution indicated the existence of three patterns of collective motility shown schematically in Fig. 7D, F: in 'Zone A' cells don't have a preferred direction, in 'Zone B' cells move in opposite directions (~135° and 315°), and in 'Zone C' all cells move with a predominant preferred direction (~45°) (Fig. 7D, F). We named these patterns swarm (Zone A), stream (Zone B), or flock (Zone C) (Fig. 7G). They were classified by likelihood analysis: the distribution of the angle velocity is constant in a swarm (all angle velocity are equally probable), bi-modal in a stream (cells are moving in equal but opposite directions), and uni-modal in a flock (cells move in one direction) (Fig. 7H). These patterns were observed in all tumor slices examined (Supplementary Figs. 18, 19, and 20). Average cell speeds differed among the three patterns (Fig. 7E and Supplementary Figs. 18, 19, and 20). In the tumor core, swarms moved faster and without orientation, followed by directionally moving flocks and streams (Supplementary Fig. 27). To determine which of these collective motion patterns match oncostream histological features, we analyzed H&E sections corresponding to imaged organotypic slices (Supplementary Fig. 16B). Cells within histological areas corresponding to streams and flocks have an aspect ratio of 2.2 and 2.7, respectively, (spindle-like cells), while those within areas corresponding to swarms have an aspect ratio of 1.2 (round cells) (Supplementary Fig. 16C, D). Moreover, elongated cells within streams and flocks are nematically aligned with each other, whereas round cells within swarms are not (Supplementary Fig. 16E). As predicted by our in silico model[48], these results suggest that cell shape, or eccentricity, is driving feature in the organization of collective motion patterns (Supplementary Fig. 16F). Therefore, taking into account cell shape and alignment, we define oncostreams as the histological expression of collective motion patterns (streams and flocks). Notice that only the dynamic analysis of collective motion can differentiate between streams and flocks. At the histological level both appear as oncostreams.

In collective motion of flocks, interactions among individual cells are sufficient to propagate order throughout a large population of starlings[49]. To define if oncostream migration patterns recall organized collective motion behavior, we analyzed the organization of the cells by performing local pair-wise correlation analysis (relative position and pair directional correlation) by tumor zones (Supplementary Fig. 17A–C). These analyses indicate the spatial correlation of location and alignment between individual cells. We observed that within swarms cells are more separated, as neighbors are located at 20–40 μm.

Streams and flocks have higher cell density, and the nearest neighbors are closer, at 20–30 μm (Supplementary Figs. 17E and 18, 19, 20). Pair-wise directional correlation with nearby neighbors showed that cell movement is positively correlated in all patterns at distances between 10 and 50 μm, with higher correlation left-to-right for streams (≈0.2), left-to-right/front-to-back for flocks (≈0.2–0.4), and a lower correlation for swarms (≈0.1) (Supplementary Figs. 17F and 18, 19, 20). We ascertained that tumor cells within oncostreams migrate in a directional manner (streams (↑↓) and flocks (↑↑)), while non-oncostream cre more separated, as neighborells move randomly without directional alignment as swarms. Thus, our analyses strongly indicate that within the tumor core of high grade glioma cells are dynamically heterogeneous and display organized collective migratory behavior associated with tumor histological and genetic features.

**Oncostreams increase the intratumoral spread of tumoral and non-tumoral cells.** Pair-wise correlation analysis showed that oncostream glioma cells are collectively organized. To test the underlying nature of collective oncostream motility, we analyzed adherent junction markers. Tumors with oncostreams were negative for E-cadherin, whereas N-cadherin was strongly expressed (Supplementary Fig. 21A), suggesting that these fascicles move in a manner akin to collective migration of mesenchymal cells of the neural crest[33,50]. Although, no difference in N-cadherin were found within oncostreams and the surrounding areas, N-cadherin was elevated in TCGA-GBM (Grade IV) tumors compared to TCGA-LGG (Grade III and II). High levels of N-cadherin correlate with lower survival in HGG patients and mesenchymal transformation (Supplementary Fig. 21B, C).

On the other hand, oncostream growth and motility is unlikely to be due to glioma proliferation. BrdU staining showed no differences between oncostream and non-oncostream regions, and in the oncostreams, the mitotic plane was always perpendicular to the main axis as expected (Supplementary Fig. 21D, E). These results are also supported by the RNA-Seq data of dissected oncostreams, where proliferation genes were not differentially expressed (Fig. 4A–C).

Collective motion could affect the distribution of other cells within the tumor. Since oncostreams are heterogeneous, we inquired about their pro-tumoral role by potentially spreading cells throughout the tumor. We designed co-implantation experiments using human glioma stem cells (MSP-12), and highly aggressive and oncostream-forming glioma cells (GL26)

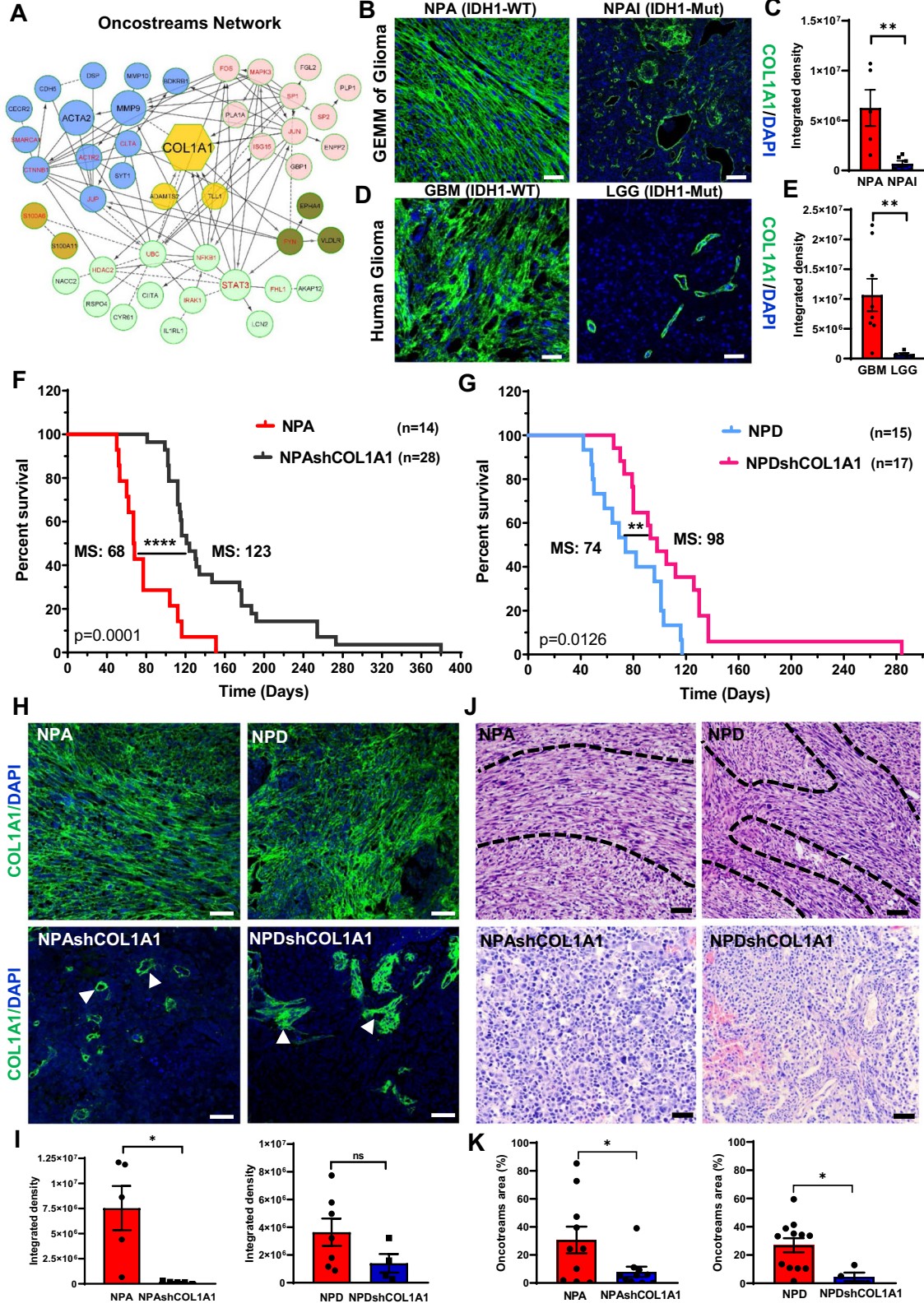

co-implanted into immunosuppressed mice. Implantation of MSP-12 cells alone generated slow-growing tumors (median survival of 6–8 months). At 21 days post-implantation, MSP-12 cells remained restricted to the injection area with an average distance of $28.9 \pm 7.73\,\mu m$ from the actual injection site. Surprisingly, when MSP-12 cells were co-implanted with GL26-citrine cells, MSP-12 cells spread throughout the tumor, moving

along oncostreams to much longer distances ($83.7 \pm 23.74\,\mu m$) from the injection site (Fig. 7I–K). Cellular cytoplasmic processes from MSP-12 cells implanted alone displayed a random distribution. However, in co-implanted tumors, such processes from MSP-12 cells are completely aligned with glioma GL26 cells within oncostreams (Fig. 7K, L and Supplementary Fig. 21F). These results strongly suggest that oncostreams function as

**Fig. 5 COL1A1 is a central hub of oncostream organization and glioma malignancy. A** Network analysis of differentially expressed (DE) genes comparing oncostreams vs. no-oncostreams. Node size reflects the degree of connectivity. Clusters of nodes with the same color illustrate modules of highly interacting genes. **B** Immunofluorescence analysis of COL1A1 expression in GEMM of glioma comparing NPA (IDH1-WT) vs NPAI (IDH1mut). Representative confocal images display COL1A1 expression in green (Alexa 488) and nuclei in blue (DAPI). Scale bar: 50 μm. **C** Bar graphs represent COL1A1 quantification. NPA $n = 5$ and NPAI $n = 6$ tumors for each experimental condition were used for the analysis. Ten fields from each tumor section were selected at random. Graphs present mean ± SEM. Two-sided t-test, **$p = 0.0087$. **D** Immunofluorescence analysis of COL1A1 expression in human GBM and LGG tumors. COL1A1 expression in green and nuclei in blue. Scale bar: 50 μm. **E** Bar graphs represent COL1A1 quantification $n = 5$ (LGG) and $n = 8$ (GBM) tumors were used. Ten fields of each section were randomly selected. Graphs present mean ± SEM. Mann–Whitney two-sided t-test, **$p = 0.0031$. **F, G** GEMM of glioma with COL1A1 inhibition. **F** Kaplan–Meier survival curve comparing NPA (MS: 68 days; $n$: 14) vs NPAshCOL1A1 (MS: 123 days; $n = 28$). **G** Kaplan–Meier survival curve comparing NPD (MS: 74 days; $n = 15$) vs. NPDshCOL1A1 (MS: 98 days; $n = 17$). Log-rank (Mantel–Cox) test. ****$p < 0.0001$, **$p = 0.0126$. **H** Immunofluorescent COL1A1 expression in NPA and NPD and *Col1a1* downregulation (NPAshCOL1A1 and NPDshCOL1A1); confocal images of COL1A1 expression in green and nuclei in blue. Arrows indicate COL1A1 enriched perivascular cells. Scale bar: 50 μm. **I** COL1A1 quantification of fluorescence integrated density. Ten fields of each tumor section were selected at random (NPA: $n = 5$, NPAshCOL1A1: $n = 5$, NPD; $n = 7$, NPDshCOL1A1: $n = 5$). Two-sided t-test, $p = 0.0104$, ns = not significant. Graphs present mean ± SEM. **J** Representative Hematoxylin and Eosin images of the histopathological identification of oncostreams comparing COL1A1 knockdown tumors versus controls. Scale bars: 20 μm. **K** Quantitative analysis of oncostream areas using deep learning analysis (NPA: $n = 10$, NPAshCOL1A1: $n = 10$, NPD; $n = 12$, NPDshCOL1A1: $n = 4$ samples used per experimental condition). Graphs present mean ± SEM; Two-sided t-test analysis, NPA vs NPAshCOL1A1 *$p = 0.0199$; NPD vs NPDshCOL1A1 *$p = 0.0124$.

intra-tumoral highways facilitating the rapid distribution of slow-moving glioma cells and/or non-tumor cells throughout the tumor mass. These findings could help explain the dispersal and intratumoral mixing of diverse clonal populations as demonstrated in previous studies, supporting an important potential role of oncostreams in determining spatial cellular heterogeneity.

**Dynamic interactions at the tumor invasive border: Oncostreams foster glioma aggressiveness through collective migration.** Furthermore, we asked whether oncostreams contribute to glioma malignant behavior. The analysis of histological sections showed that multicellular fascicles of elongated and aligned cells are found invading from the tumor border into the normal brain parenchyma (Supplementary Fig. 22A). The formation of streams around blood vessels was also observed (Supplementary Fig. 22A). These patterns of invasion are also detected using our deep learning methods (Supplementary Fig. 22B).

We then used our glioma explant model to analyze the glioma dynamics by time-lapse confocal imaging at the tumor invasive border (Figs. 8A and Supplementary Fig. 23). We implanted glioma NPA GFP+ cells into tdTomato (mT/mG) mice so the border between tumor cells and normal brain could be delineated easily. We observed that glioma cells that extended from the tumor border to the normal brain parenchyma adopted different dynamic patterns. At the invasive tumor border cells were seen moving as isolated cells, or as collective migratory fascicles that resembled oncostream structures found in the tumor core (Fig. 8B–G and Supplementary Figs. 23–26).

To objectively distinguish between different dynamic patterns, we determined the angle velocity distribution, and the likelihood that distributions corresponded to either a stream, a flock, or a swarm. We found streams along the perivascular niche, and invading brain parenchyma without following any pre-existing brain structures, as well as cells invading as flocks, and swarms (Fig. 8E, F and Supplementary Figs. 23D, G, H and 24–26A, C, D). The correlation of position and pairwise correlation supports the existence of invading collective motion structures in NPA tumors with high expression of COL1A1 (Supplementary Fig. 22D, E and Supplementary Figs. 23J, K and 24–26F, G). We also determined the participation of collagen fibers in oncostreams migration Immunofluorescence analysis on explant slices showed that collagen fibers are aligned along multicellular fascicles of glioma cells invading the normal brain. These data show how

collagen fibers serve as scaffolds for collective tumoral cell invasion (Supplementary Fig. 27).

Our data indicate the existence of a complex framework of collective motion patterns at the glioma border, that is consistent with previous descriptions[34]. Although the patterns observed at the NPA tumor border are similar to those of the tumor core, cell speed differed between the areas. Cells in the tumor core displayed significantly lower average speeds (stream: 4.26; flock: 5.95, swarm: 6.27 μm/h) compared to cells at the tumor invasive border (stream: 7.95; flock: 7.55, swarm: 8.01 μm/h) (Supplementary Fig. 28A, B).

Then, we asked whether the knockdown of *Col1a1* in NPA gliomas affects changes in the patterns of migration. Analysis of tumor cells (GFP+) at the tumor borders of GEMM of gliomas comparing NPA and NPA-shCOL1A1 showed a difference in the cellular patterns found at the tumor invasive border. The analysis of tumor borders revealed an increase in the sinuosity of NPA tumors, a finding compatible with NPA tumors exhibiting a higher proportion of collective migration into the normal brain when compared to NPA-shCOL1A1 tumors (Fig. 8G–I and Supplementary Fig. 29).

Moreover, the time lapse-confocal imaging and migration analysis of NPA-shCOL1A1 explants showed that in these tumors, cells invade the normal brain parenchyma as isolated cells (Supplementary Figs. 30–33). Velocity angle, velocity vector and the likelihood analysis indicated that the overall distribution corresponded predominantly to swarm random patterns (Supplementary Figs. 30–33D, E, F). Further analysis of Relative Position Correlation and Pairwise correlation supports the presence of low density of cells compatible with single cell invasion patterns in NPAshCOL1A1 tumors with low expression of collagen (Supplementary Figs. 30–33G, H).

We conclude that oncostreams (streams and flocks) are organized collective migratory structures enriched in COL1A1 that participate in the dynamic organization of the tumor microenvironment within the tumor core and at the tumor invasive border of high-grade gliomas, impacting the malignant behavior of gliomas. Depletion of collagen1A1 eliminates oncostreams and their associated functions.

**Intravital imaging of glioma reveals the existence of oncostreams' collective motion patterns in vivo.** To determine whether our previously described collective migration patterns of glioma cells ex vivo occur also in vivo we performed high resolution time lapse intravital imaging using two photon microscopy.

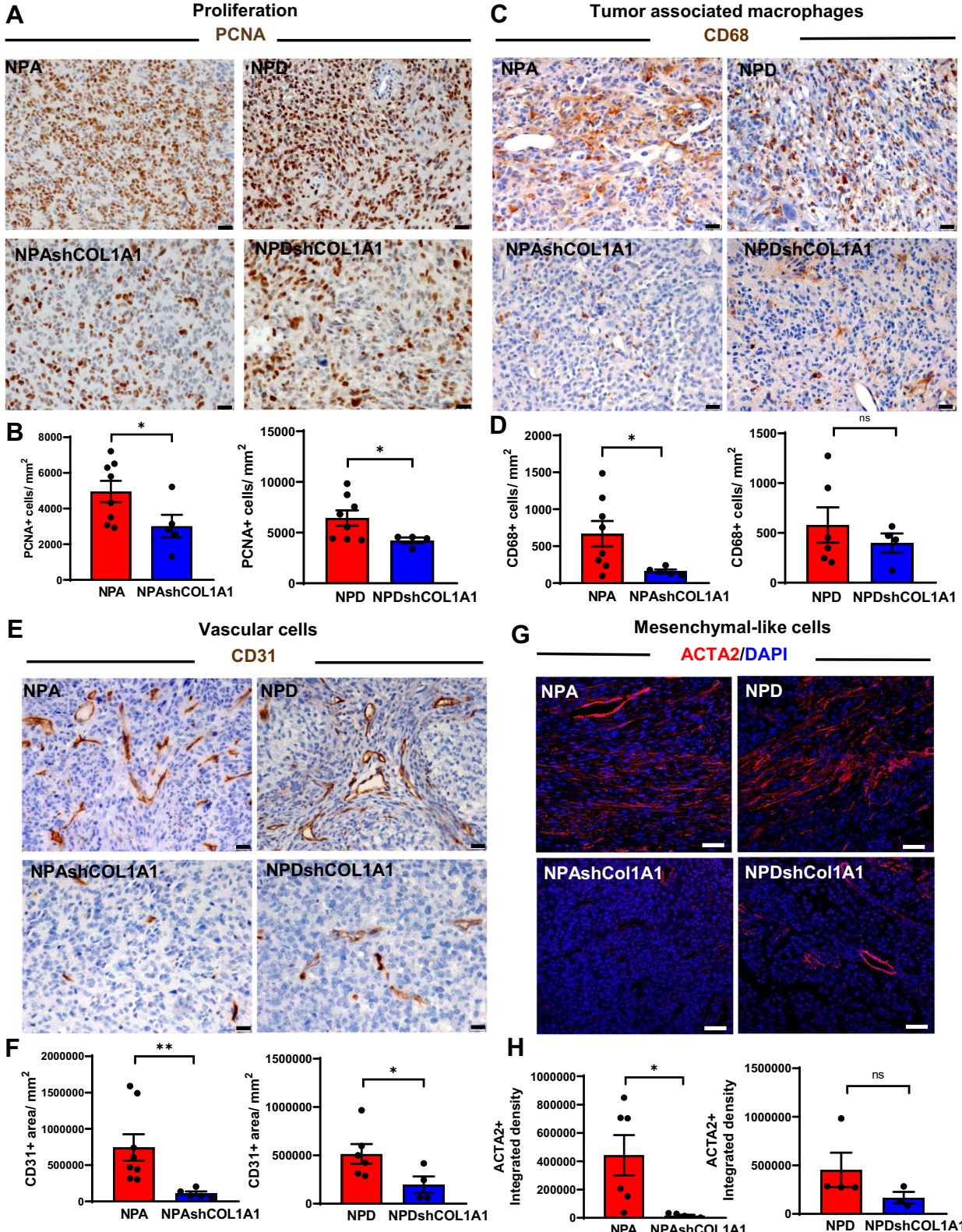

To do so NPA glioma cells were intracranially implanted in the brains of tdTomato (mT/mG) mice at a depth of 0.8 mm (Fig. 9A). To visualize cell migration, we established a cranial window above the injection site (Fig. 9B). After 7–15 days of tumor growth, we acquired z-stack images to obtain a 3D orthogonal view of the tumor growing within the cortex. This allowed us to establish intravital imaging below the brain surface.

Next, we selected a position at a depth of >100 μm and proceeded to acquire time lapse images of the tumor growing in the normal parenchyma at an interval of 5 min, for 8–12 h (Fig. 9C, D and Supplementary Figs. 35A, 36A, B, and 37A, B).

In some cases, to determine the exact anatomical location of the tumor, following intravital imaging, we perfused-fixed the brain and performed fluorescence immuno-histochemistry

**Fig. 6 Knockdown of COL1A1 within glioma cells modifies the tumor microenvironment.** Immunohistochemical analysis (**A**, **C**, and **E**) of GEMM of glioma controls (NPA and NPD) and COL1A1 downregulation (NPAshCOL1A1 and NPDshCOL1A1). **A** Representative images of PCNA expression. Scale bar: 20 μm. **B** Bar graphs represent the quantification of PCNA+ cells numbers (cells/mm$^2$) using QuPath positive cell detection. Graphs present mean ± SEM, (NPA: $n = 8$, NPAshCOL1A1: $n = 5$, NPD: n = 8, NPDshCOL1A1: $n = 4$), two-sided t-test with Welch's correction, *$p = 0.0240$. **C** Representative images of CD68 expression. Scale bar: 20 μm. **D** Bar graphs represent CD68+ cell quantification (cells/mm$^2$) using QuPath positive cell detection. Graphs present mean ± SEM, (NPA: $n = 8$, NPAshCOL1A1: $n = 5$, NPD: $n = 6$, NPDshCOL1A1: $n = 4$), Two-sided t-test, *$p < 0.05$, ns: no significant. **E** Representative images of CD31 expression. Scale bar: 20 μm. **F** Bar graphs represent CD31 + cells quantification (cells/mm$^2$) using QuPath positive cells detection. Error bars represent ± SEM, (NPA: $n = 8$, NPAshCOL1A1: $n = 5$, NPD: $n = 6$, NPDshCOL1A1: $n = 4$). Graphs present mean ± SEM. Mann–Whitney two-sided t-test, NPA vs NPAshCOL1A1 **$p = 0.0016$, NPD vs NPDshCOL1A1 *$p = 0.0381$. **G** Immunofluorescence analysis of GEMM of glioma controls (NPA and NPD) and COL1A1 downregulation (NPAshCOL1A1 and NPDshCOL1A1). Representative images of ACTA2 expression in red (Alexa 555) and nuclei in blue (DAPI). Scale bar: 50 μm. **H** Bar graphs represent ACTA2 quantification in terms of fluorescence integrated density. Graphs present mean ± SEM, (NPA: $n = 6$, NPAshCOL1A1: $n = 5$, NPD: $n = 4$, NPDshCOL1A1: $n = 3$), two-sided t-test, *$p = 0.0242$, ns: no significant.

analysis on paraffin-embedded coronal sections which were imaged by confocal microscopy (Supplementary Fig. 35B, C). The location of the area imaged by intravital microscopy (>100 μm) is shown. In this figure, we can identify the location of the tumor (GFP+ cells) containing parenchymal blood vessels (TdTomato+), and astrocytic processes (GFAP+) (Supplementary Fig. 35B, C). Thus, together with Supplementary Fig. 35A, we demonstrate imaging below the brain surface and within the normal parenchyma of the brain.

To analyze the movement of GFP+ glioma cells we used Fiji's plug-in Track-Mate. In Fig. 9 the imaged area of the tumor core was divided into subregions to determine the existence of migration patterns (Fig. 9E, F). The analysis of cell migration in vivo showed that the glioma cells exhibit organized, nematically aligned cells moving collectively at a depth of 120 μm. Angle velocity distribution analysis determined the existence of stream collective motion patterns in the three delimited subregions (for example in Fig. 9H), illustrating that aligned cells are moving in opposite directions.

To corroborate the existence of stream patterns we applied our likelihood analysis. For streams, the distribution of the angle velocity and velocity vectors displayed a bi-modal distribution (cells were moving in equal but opposite directions) (Fig. 9I), similar to that observed in the explant models. Mean speed varied from 5.50 to 9.95 μm/h (Fig. 9G). A different tumor imaged at a depth of 145 μm also showed the presence of streams (Supplementary Fig. 36).

We then analyzed the tumor invasive border in contact with normal brain (Supplementary Fig. 35). At an imaging depth of 140 μm, cells at the tumor border displayed stream collective dynamics (Supplementary Fig. 35D, Zone A and B). These motion patterns were determined using the angle velocity distribution analysis, and likelihood distribution analysis (Supplementary Fig. 35G, H). Further analysis of Relative Position and Pairwise correlation supports the presence of high density of cells compatible with collective migration patterns (Supplementary Figs. 34B, C, 35I, J, and 36I, J).

To determine whether the knockdown of *Col1a1* in NPA gliomas alters the migration patterns observed for NPA gliomas we analyzed a NPAshCOL1A1 tumor by two-photon microscopy at a depth of 110 μm (Supplementary Fig. 37A, B).

The intravital migration analysis of NPAshCOL1A1 gliomas showed that glioma cells migrate without a preferred direction, as swarms (Supplementary Fig. 37C–G). The alignment analysis, Relative Position, and Pairwise correlation confirm the presence of not-aligned low-density cells compatible with single cell invasion patterns in gliomas with *Col1a1* downregulation (Supplementary Fig. 37E, J, K).

The collective motion patterns found in vivo resemble the collective motion patterns described using the ex vivo explant model. Our results show that glioma cells expressing collagen are

organized in collective dynamic patterns at the tumor core and the tumor invasive border, in tumor explants, and in in vivo intravital models of gliomas analyzed by two photon microscopy.

## Discussion

Mesenchymal transformation is a hallmark of tumor heterogeneity that is associated with a more aggressive phenotype and therapeutic resistance[13,18,21]. Mesenchymal transformation involves fibroblast-like morphological changes associated with active migration and gain of expression of mesenchymal genes as previously described[21,22].

Herein we present a comprehensive study that defines the morphological, cellular, dynamic, and molecular properties of multicellular mesenchymal-like structures within gliomas. These structures are fascicles of aligned spindle-like cells found throughout the tumors and represent areas of mesenchymal transformation. We interpret these structures to be the histological expression of areas of collective motion of glioma cells, and we refer to them as oncostreams.

Oncostreams are areas of mesenchymal transformation and are identified histologically as fascicles of aligned and elongated cells. When examined dynamically, we found that tumor cells move by collective motion within the tumor core and at the invading border. The capacity to identify areas of collective motion in histological sections has allowed us to characterize the molecular organization of such dynamic structures. We thus describe the overall molecular mechanisms that govern the organization and function of these structures and demonstrate the causal role of individual mediators. Surprisingly, we discovered that COL1A1 is central to the structural and dynamic characteristics of oncostreams. Indeed, the loss of *Col1a1* expression from tumor cells disrupts the structural and functional characteristics of oncostreams, resulting in a complete loss of mesenchymal areas within gliomas and a reduction in glioma malignant behavior (Fig. 10).

The analysis of the gene ontologies over-represented within oncostreams indicates that oncostreams denote areas enriched for "positive regulation of cell migration", and in mesenchymal-related genes. Interestingly, *Col1a1* appeared as a central hub of oncostream organization and mesenchymal transformation. We postulate that oncostreams are the histopathological expression of patterns of collective motion (i.e., streams and flocks) in high grade glioma tumors. Different strategies of cell migration encountered in our gliomas are reminiscent of migratory characteristics observed during embryonic development[31,32,50]. In developmental biology, collective motion is represented by cells moving together in clusters, sheets, streams, or other multicellular arrangements[28,31,32].

Our studies of oncostream dynamics at the tumor core are compatible with the results of Ralitza et al.[51]. This group studied ex vivo explant slices of spontaneous intestinal carcinoma, and showed that cells within the tumor core were highly dynamic and

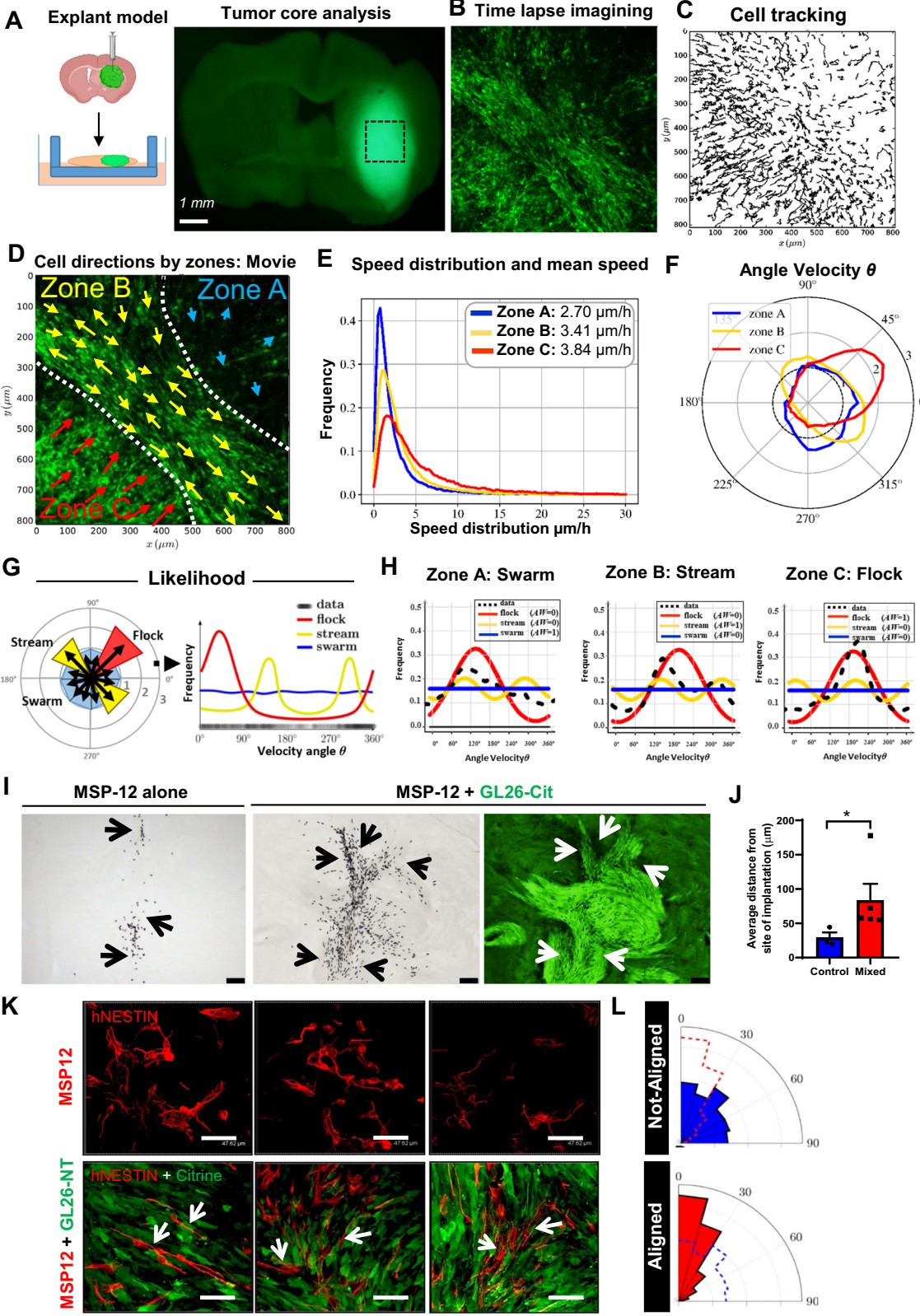

**A** Explant model    Tumor core analysis

**B** Time lapse imagining

**C** Cell tracking

**D** Cell directions by zones: Movie

Zone B    Zone A

Zone C

**E** Speed distribution and mean speed

Zone A: 2.70 μm/h
Zone B: 3.41 μm/h
Zone C: 3.84 μm/h

**F** Angle Velocity θ

zone A
zone B
zone C

**G** Likelihood

Stream    Flock

Swarm

data
flock
stream
swarm

**H** Zone A: Swarm    Zone B: Stream    Zone C: Flock

**I** MSP-12 alone    MSP-12 + GL26-Cit

**J**

**K** hNESTIN    hNESTIN + Citrine    MSP12    MSP12 + GL26-NT

**L** Not-Aligned    Aligned

display directionally correlated cell motion[51], similar to our results described herein. Recent in silico based mathematical modeling of glioma cell dynamics by our group, showed that only elongated cells, but not spherical cells, are able to form organized aligned cellular structures in a cell-density dependent manner[48]. Our modeling studies strongly support our in vivo and ex vivo data described in this manuscript.

Moreover, it has been described that increased matrix cross-linking, enzymatic remodeling, and parallel orientation of matrix collagen fibers stiffens tissue, modifies cell morphology, and promotes cell migration and invasion[36,39,40,52,53]. Our results support the proposal that oncostreams serve as highways to spread tumor, and non-tumor cells, throughout the tumor. Indeed, oncostream fascicles contain higher amounts of

**Fig. 7 Collective dynamics of oncostreams increase cell spreading within the tumor core. A** Experimental setup: NPA-GFP glioma cells implantation and explant slice cultures generation for confocal time-lapse imaging. (Created in BioRender.com). **B** Single representative time-lapse confocal image of glioma cells within the tumor core (Movie #1). Tumor cells are GFP+ (green). **C** Tracking analysis of individual cell paths performed using the Track-Mate. **D** Preferred directions of cells within three zones (A–C) superimposed onto a representative time lapse-image. **E** Speed distribution and mean speed (μm/h) in Zones A, B, and C. **F** Distribution of angle velocity (θ) for each zone. The plot shows the proportion of cells moving in angle direction θ for each zone. **G**, **H** Classification of collective motion patterns: stream (two peaks = 2 preferred angle velocity), flock (only one peak) or swarm (no preferred angle velocity). **G** Angle Velocity was transformed to a histogram; these data were then used to calculate the likelihood that corresponds to either a stream, flock, or swarm. The results are given in (**H**) for each zone. The frequency distribution of the data (shown in black) uses a non-parametric estimation (kernel density estimator). We tested three types of distributions, ρ, to describe the dataset and give a likelihood for each case. The best fit was then determined by the Akaike weight (AW). **I** Co-implantation of highly malignant GL26-citrine cells (green) and human MSP-12 glioma cells (ratio 1:30), and MSP-12 cells alone (control – left image). Immunohistochemistry of human nuclei (black) denotes MSP-12 cells. Arrows show the distribution of MSP-12 cells. Scale bar: 100 μm. **J** Quantification of the distance of MSP-12 from the site of implantation. Biologically independent tumors, n = 3 for control and n = 5 (MSP-12 + GL26). Error bars ± SEM; Mann–Whitney two-sided t-test, *p = 0.0357. **K** Immunofluorescence images of human-nestin (red) labeling MSP-12 cells, and GL26-citrine cells. Experiment was performed in three independent tumor samples. Scale bar: 47.62 μm. **L** Angle histogram plots quantify the alignment of MSP-12 within oncostreams, and the random alignment of MSP-12 cells when implanted alone (with dashed overlays of the other condition's alignment).

---

macrophages/microglia and mesenchymal cells. Dispersal of tumor and non-tumoral cells throughout the tumors could help explain the mixing of different clonal populations seen in molecular studies of high grade gliomas[10].

This study contributes to explaining how a particular feature of intratumoral heterogeneity, namely mesenchymal transformation, affects HGG progression. Our data indicate that the density of oncostreams plays a potential role in overall glioma malignant behavior in mouse and human gliomas.

Spatially resolved transcriptional analysis using LCM provided insights into the molecular mechanisms that regulate oncostream functions. Oncostreams were defined by a specific transcriptomic signature that matched our immunohistochemical studies. *Col1a1* overexpression within oncostreams was complemented with the overexpression of extracellular matrix proteins such as *Mmp9, Mmp10, Adamts2*, which are known to remodel and participate in the reorganization of collagen fibers. Oncostream fascicles were correspondingly enriched in *Col1a1* when assessed by immunohistochemistry.

Within the extracellular matrix, collagen fibers would be able to constitute a scaffold for the organization of the tumor microenvironment and thus potentially act to promote tumor infiltration and invasion. While collagen was previously thought to be a passive barrier that could reduce tumor invasion, it has now been shown that collagen fibers can serve as mechanical and biochemical tracks that facilitate cellular migration and tumor progression[36,37,39,54]. Previously, multi-cancer computational analysis found that within a mesenchymal transformation signature in different cancers including gliomas, *COL1A1* was one of the top differentially expressed genes[18,22,55]. COL1A1 is overexpressed in high grade malignant gliomas and its expression levels are inversely correlated with patient survival[56] as indicated in https://www.cancer.gov/tcga. In our mouse glioma models and in human gliomas, tumors with higher density of oncostreams also express higher levels of COL1A1. COL1A1 is a consistently differentially expressed gene in the glioma mesenchymal signature identified in malignant gliomas and in glioma stem cells as described in previous studies[4,9,57]. Overall, our data are in agreement with a recent study by Puchalski et al., which assigned genetic and transcriptional information to the most common morphological hallmarks of a glioma, emphasizing the importance of integrative histo-molecular studies[8].

Surprisingly, our data indicate significant plasticity of the mesenchymal phenotype in gliomas, similar to other studies[13,15]. Genetic inhibition of *Col1a1* within glioma cells depleted of *Col1a1* from tumors, eliminated oncostream structures, reduced the glioma malignant phenotype, and prolonged animal survival.

Our findings are comparable with results from various studies that investigated the in vitro and in vivo consequences of collagen depletion, inhibition of collagen cross-linking, or collagen synthesis inhibition on normalizing tumor ECM. In these studies, inhibition of collagen led to changes in the ECM which improved drug penetration, efficacy, as well as tumor access of therapeutic nano-particles or gene-based therapies[58–62]. In addition, *Col1a1* inhibition within glioma cells induced cell intrinsic and extrinsic changes in the TME. *Col1a1* inhibition not only inhibits tumor cell proliferation and migration but also decreased the infiltration of microglia/macrophages, endothelial cells proliferation, and perivascular mesenchymal-like cells. As previously shown by other studies, glioblastomas exhibit complex interactions between tumoral and non-tumoral cells, (including macrophages, immune cells, and endothelial cells), which influence tumor growth, transformation, invasion, and response to treatment[9,13,63]. However, a major remodeling of the tumor mesenchymal phenotype in response to inhibition of *Col1a1* has not been described earlier.

Moreover, we found that multicellular oncostream fascicles are detected in both ex vivo and in vivo glioma models, and that oncostreams may potentially act to facilitate tumor cell invasion, thereby increasing glioma aggressiveness. Our findings strongly support the importance of collective motility of glioma cells in the progression of tumor growth and possibly the invasion of normal brain parenchyma, and is supported by earlier studies of normal and pathological conditions[27,34,35,64–68]. Several lines of evidence suggest that oncostream fascicles within the tumor core are interspersed with normal brain cells, i.e. GFAP+ astrocyte processes, and positive neurofilament staining; in addition, tumors growing in TdTomato mice show green tumor cells intermingled with normal brain (shown by its red fluorescence). However, we acknowledge the limitations of intravital imaging in evaluating the interactions between tumor cells and individual normal brain cells and the significance of our data for the migratory behavior of human gliomas. Further studies using transgenic mice expressing fluorescent proteins in normal brain cells would be valuable to understand oncostream interactions with individual normal brain cells in the tumor core, and at the tumor borders. Although oncostreams are present throughout the tumor core, and tumor invasive borders, the detailed interactions of oncostreams with normal brain awaits further experiments and technical developments; we recognize that the current experiments do not address diffuse brain infiltration in human gliomas. Additional strategies, using fresh human explant models will be necessary to expand our study and characterize oncostream migration and interaction with the normal brain parenchyma.

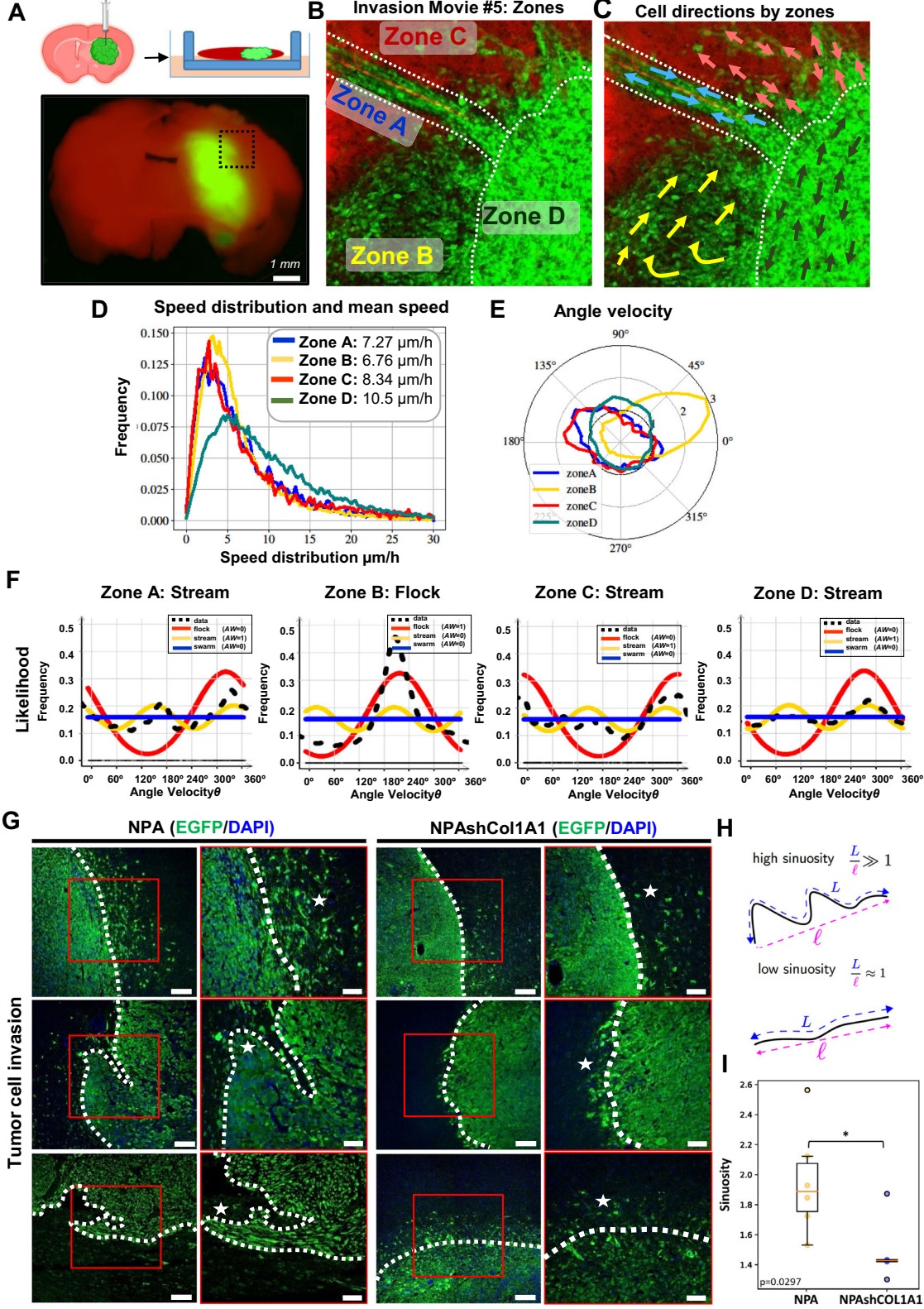

In summary, our observations suggest that oncostreams are morphologically and molecularly distinct structures that represent areas of collective motion that contribute to tumor growth and aggressiveness. These malignant dynamic structures overexpress *Col1a1*. *Col1a1* knockdown eliminates oncostreams, reduces the mesenchymal phenotype, modifies the TME, and slows tumor progression. Our findings provide alternative views towards understanding tumor mesenchymal transformation and its therapeutic treatment. We propose that depletion of *Col1a1* within glioma cells is a promising approach to reprogram mesenchymal transformation in glioma tumors, and could be harnessed as a therapeutic approach, and reduce the glioma malignant phenotype.

**Fig. 8 Collective migration of COL1A1 enriched oncostreams contributes to malignant glioma behavior at the tumor invasive border. A** Experimental setup and location of imaging of tumor borders using td/mtTomato mice (Movie #5) (Created in BioRender.com). **B** Representative time-lapse scanning confocal image of glioma cells at the tumor border and subdivision into different zones. The image shows tumor cells (green) and brain parenchyma (red). **C** Preferred direction of cells within different zones superimposed onto a representative time lapse-image. **D** Histogram of speed distribution and mean speed ($\mu m/h$) of Zones A, B, C, and D. **E** Angle Velocity distribution ($\theta$) analysis performed by zones. Plot shows overall direction and magnitude of cell movement. **F** Likelihood analysis of the dynamic patterns at the tumor border. Graph of density estimation $\rho$ flock (red), $\rho$ stream (yellow), and $\rho$ swarm (blue). Black line (data) uses a non-parametric estimation. AW:0 or AW:1. **G** Immunofluorescence analysis of GFP+ cells in Genetically Engineer Mouse Models of glioma controls NPA ($n = 6$), and NPAshCOL1A1 ($n = 5$). GFP is shown in green (Alexa 488) and nuclei in blue (DAPI). Dotted lines show tumor borders. Stars show tumor invasive patterns. Notice the absence of collective invasion patterns in NPAshCOL1A1. Scale bar: main(left) 100 μm, inset(right) 50 μm. **H** Tumor borders were analyzed using the Allen-Cahn equation. Images were split into two values ($-1$ and $+1$) representing the inside and outside of the tumor. Illustration of the sinuosity of a curve: it is defined as the ratio between the length of the curve L and the distance between the two extreme points. **I** Sinuosity of the border for all experiments. NPA: $n = 6$ and NPAshCOL1A1: $n = 5$ tumors for each experimental condition were used for the analysis. We detected a decrease of the sinuosity in COL1A1 knockdown tumors. t-test unequal variance, *$p = 0.0297$. Center of the boxplot is the median value, upper bound of the boxplot is the Q3 (75th percentile), the lower bound of the boxplot is the Q1 (25th percentile), the upper whiskers is the maximum value of the data (within 1.5 times the interquartile range over the 75th percentile), and the lower whisker is the minimum value of the data (within 1.5 times the interquartile range under the 25th percentile).

## Methods

**Glioma cell lines and culture conditions**. Mouse glioma cells (NPA, NPD, NPAshCOL1A1, NPDshCOL1A1, and GL26) and human glioma cells (MSP-12, SJGBM2) were maintained at 37 °C with 5% $CO_2$ and their respective media[42–45]. Mouse NPA, NPD, NPAshCOL1A1, NPDshCOL1A1 neurospheres were derived from our genetically engineered tumors using the Sleeping Beauty (SB) transposase system as[42–45]. Mouse GL26 glioma cells were generated by Sugiura K and obtained from the frozen stock maintained by the National Cancer Institute (Bethesda, MD)[23]. MSP-12 human glioma cell lines were provided by Christine Brown, City of Hope, and SJGBM2 human glioma cells were provided by Children's Oncology Group (COG) Repository, Health Science Center, Texas Tech University.

**Intracranial implantable syngeneic mouse gliomas**. All in vivo experiments were conducted according to the guidelines approved by the Institutional Animal Care and Use Committee (IACUC) at the University of Michigan protocols PRO00009599, PRO00009578, and PRO00009551. All animals were housed in an AAALAC accredited animal facility at the University of Michigan. Animals were monitored daily. All tumor bearing animals were euthanized at the time they develop clinical signs of tumor burden (impaired mobility, ruffled fur, hunched posture, protoporphyrin staining around eyes and nose, hunched posture, loss of coordination). Maximal tumor burden did not exceed IACUC guidelines.

Glioma tumors were generated by stereotactic intracranial implantation into the mouse striatum of $3.0 \times 10^4$ mouse glioma cells (either, NPA, NPD, or GL26) in 6–8 week old females C57BL/6 mice (Taconic Biosciences) or human glioma cells (SJGBM2) in immune-deficient 6–8 week old males NSG mice, following our methodology[43–45,69]. To test whether oncostream tumor cells help move other cells throughout the tumor we generated a co-implantation glioma model by intracranial implantation of highly malignant GL26-citrine cells with low aggressive human MSP12 glioma cells at a ratio of 1:30 (1000 GL26-citrine cells and 30,000 MSP12 cells) in immune-deficient NSG mice. As controls, NSG mice were implanted with 30,000 MSP12 cells alone or 1000 GL26-citrine cells alone as controls.

**Genetically engineered mouse glioma models (GEMM)**. We used genetically engineered mouse glioma models for survival analysis and histopathological analysis. Murine glioma tumors harboring different genetic drivers were generated using the Sleeping Beauty (SB) transposon system[42–45]. Genetic modifications were induced in postnatal day 1 (P01) male and female wild-type C57BL/6 mice (Jackson Laboratory), according to IACUC regulations.

To design and clone the shRNA targeting the *Col1a1* gene (pT2-shCol1a1-GFP4), we tested 3 22-base pair sequences for the mouse *Col1a1* gene selected from candidate sequences within the RNAi codex database (http://cancan.cshl.edu/cgi-bin/Codex/Codex.cgi) and InvivoGene (http://www.invivogen.com/sirnawizard). shCol1a1-A: CCCTGGTGATACTGGTGTTAAA, shCol1a1-B: GCAACAGTCG CTTCACCTACA, shCol1a1-C: GCAAGACAGTCATCGAATACA and shRNA Scramble: GAATCTAATCGTATCGTGGCTG. To confirm *Col1a1* knockdown, NIH/3T3 mouse cells were transfected with the designed shRNA plasmids for 72 h and subsequently analyzed by Western Blot[43]. COL1A1 protein levels were normalized to β-ACTIN levels and compared to the Scramble control. Uncropped membrane scans are supplied in the Source data file.

Genetic models generated include the following gene expression/inhibitions: (i) *shP53*, *NRAS-G12V* and *shAtrx* (NPA), (ii) *shP53*, *NRAS-G12V*, *shAtrx* and *IDH1-R132H* (NPAI), (iii) *shP53*, *NRAS-G12V* and *Pdgfβ* (NPD), (iv) *shP53*, *NRAS-G12V*, *shAtrx*, *shCol1a1* (NPAshCOL1A1), (v) shP53, *NRAS-G12V*, Pdgfβ, *shCol1a1* (NPDshCOL1A1). Plasmid sequences were verified by Sanger sequencing. Plasmid sequences described below were used to generate tumors: (i) pT2C-LucPGK-SB100X, transposon & luciferase expression; (ii) pT2-NRAS-G12V,

RTK/RAS/PI3K activation; (iii) pT2-shP53-GFP4, P53 knock-down; (iv) pT2-shAtrx-GFP4, ATRX knock-down; (v) pKT-IDH1(R132H)-IRES-Katushka, mIDH1 expression; (vi) pT2-shP53-Pdgfβ-GFP4, P53 knock-down in combination with PDGFβ ligand overexpression. The pT2CAG-NRAS-G12V and pT2-shP53-GFP4 plasmids were the generous gift of Dr. John Ohlfest (University of Minnesota). Mice were injected with the plasmids. Plasmid uptake in pups and tumor growth was monitored by IVIS® Spectrum imaging. Adult mice displaying symptoms of morbidity were transcardially perfused[43].

**Analysis of human glioma tissues**. Oncostream presence was analyzed in unidentified H&E sections of paraformaldehyde-fixed paraffin-embedded human glioma samples obtained from primary surgery from the University of Michigan Medical School Hospital. All patients gave informed consent for collection of tissue collection under Institutional Review Board–approved protocols (HUM00057130 and HUM00024610) at the University of Michigan.

To determine the presence of oncostreams in a large cohort of human glioma tissues we used the biospecimens from "The Cancer Genome Atlas Research Network" (TCGA) from the Genomic Data Commons Data Portal, National Cancer Institute, NIH (https://portal.gdc.cancer.gov). We analyzed primary Glioblastoma multiforme (TCGA-GBM) and Low-Grade Glioma (TCGA-LGG) databases. We selected cases that have available the Slide Image and diagnostic Slides. The diagnostic slides are available for TCGA-GBM: 389 patients and TCGA-LGG: 491 patients. The presence of oncostreams was scored on 100 TCGA-GBM Grade IV tissue samples and 120 TCGA-LGG samples.

Oncostream presence was analyzed on tumors classified by histology grade (grade II, III, and IV) as shown in Supplementary Table 1. GBM were also analyzed by molecular subtypes: Classical (CL), Mesenchymal (MES), and Proneural (PN). LGG were classified in relation to grade and subtypes which consider IDH status and 1p/19q co-deletion (IDHmut-codel, IDHmut-non-codel and IDHwt) (Supplementary Data 1). H&E histology samples were analyzed at high magnification using the Slide Image Viewer from the data portal. Histological material containing brain tumors and oncostreams of both rodent and human gliomas were evaluated by C.G.K. (board certified pathologist), A.C. and PRL. Concordance regarding the presence or absence of oncostreams, between the three evaluators, was >90%. Due to the need for good quality morphological preservation, only PFPE sections were used. Clinical data including age, sex, pathology, survival, treatment information, and data including information of MGMT DNA methylation status, IDH1 mutation status, G-CIMP DNA methylation status, were obtained from http://firebrowse.org, http://gliovis.bioinfo.cnio.es and https://portal.gdc.cancer.gov data portals (Supplementary Data 1).

**Cell aspect ratio and alignment analysis in H&E tumor sections**. Images were obtained using bright-field microscopy of H&E stained paraffin sections (Olympus BX53 Upright Microscope). Tumors were imaged using 40× and 20× objectives. Images were processed using the program ImageJ. Image processing included color deconvolution to isolate nuclei, transforming images to 8-bit, and adjusting the threshold to remove background. Elliptical overlay masks were then imposed over the nuclei to match their shape. The shapes of these masks were then analyzed for the shape descriptors aspect ratio and circularity. For cell alignment analysis images were processed using ImageJ. Briefly, image processing included creating a ROI around the stream area and the non-stream area. These ROIs were then saved as separate images. Both images underwent color deconvolution to isolate nuclear stain, 8-bit conversion, and threshold adjustment to remove background. Images were then analyzed for the Feret's angle of the nucleus. The Feret's angle is the angle (0°–180°) that is taken from the x-axis to a line parallel with the longest distance across the particle being considered. Histograms were generated with these data in MATLAB using the Matplotlib plugin.

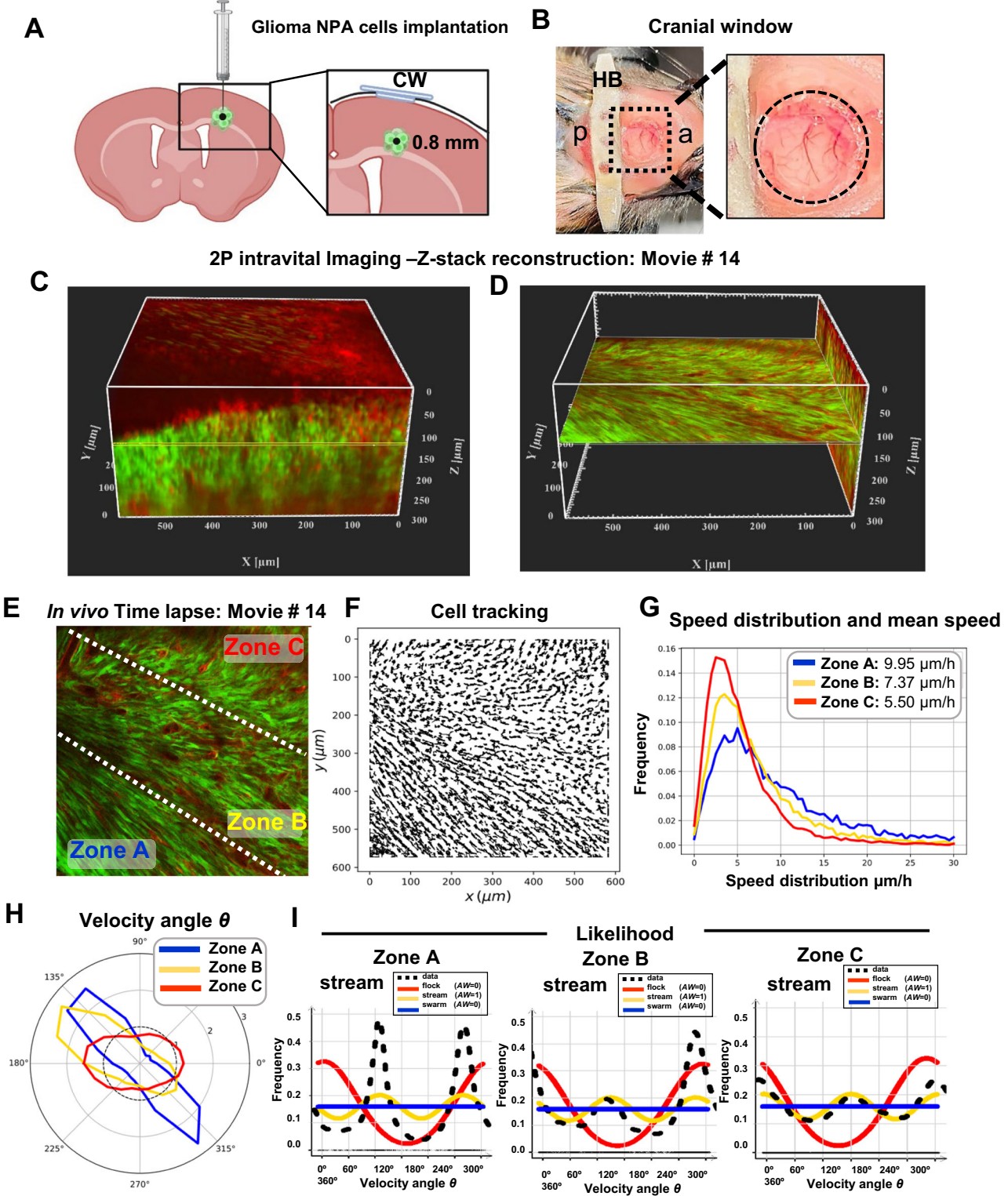

**Deep learning analysis for oncostreams detection on H&E staining of glioma tissue**. A fully convolutional neural network (fCNN) was trained in order to identify and segment oncostreams in histologic images[70]. We implemented a U-Net architecture to provide semantic segmentation of glioma specimens using deep learning[71–73]. Our oncostream dataset consisted of images from mouse tissues and open-source images from The Cancer Genome Atlas (TCGA). A total of 109 hematoxylin and eosin (H&E) stained histologic mouse images and 64 from TCGA were reviewed and oncostreams were manually segmented by the study authors (A.C, A.E.A, and P.R.L.). Images from both datasets were then augmented by randomly sampling regions within each image to generate unique patches (~300 patches/image). The location and scale of each patch was randomly chosen to allow

for oncostream segmentation to be scale invariant. First, using the mice dataset only, six iterations of training/validation set splits (80%/20%) were generated by randomly sampling unique, non-overlapping regions from each of labeled histologic images and used for model selection/hyper-parameter tuning and model testing, respectively. Our fCNN was trained for binary classification of foreground (oncostream) and background (non-oncostream) using a binary cross-entropy loss function. Both pixel-level classification accuracy and intersection over union (IOU) were used as metrics to evaluate model performance. IOU values over the testing set there were manually annotated. Our model generalized well to both mouse and human data. In addition to having a mean IOU value of 80.3%, our model did not make any false positive errors in TCGA grade II gliomas (Supplementary Table 7).

**Fig. 9 Intravital two-photon imaging reveals the collective patterns of glioma oncostream dynamics in vivo. A** Schematic representation of cell implantation for intravital two-photon imaging. The inset shows glioma cells implantation at 0.8 mm depth. CW: glass cranial window (created in BioRender.com). **B** Representative photograph of the head of an animal implanted with a cranial window showing the metallic head-bar (HB) positioned on the skull posterior to the cranial window. a: anterior; p: posterior. **C** A high-resolution 3D z-stack spanning up to 300 µm depth (starting at the brain's surface) acquired on a multiphoton microscope, imported into the Imaris. 300 x–y frames from the brain's surface were taken at a depth increment of 1 µm (voxel size = 1) at a resolution of 1024 × 1024 pixels. *XYZ* axes of the 3D image are shown in white (596 × 596 × 300 µm), and the yellow line shows the exact imaging plane for time-lapse data acquisition in vivo (at 120 µm depth). Red fluorescent protein: normal brain parenchyma. GFP+: tumor cells. **D** This panel represents the *X–Y* and the *Y–Z* plane of the reconstructed 3D image (shown in **C**) using the Orthoslicer 3D function of the Imaris viewer to illustrate the depth of the imaging plane. The *X–Y* plane shown at 120 µm depth illustrates the actual imaging position for movie #14 shown in (**E**). **E** Single representative time-lapse two-photon image of glioma cells within the tumor core (Movie #14) imaged at a depth of 120 µm, showing Zones A, B, and C. **F** Individual cell trajectories of the intravital time-lapse experiment. **G** Speed distribution and mean speed (µm/h) for Zones A, B, and C, as indicated in (**E**). **H** Angle Velocity distribution for each zone's in movie #14. The plot shows the proportion of cells moving in the angle direction θ for each zone. **I** Likelihood analysis of the dynamic patterns determined for each zone of Movie #14. The frequency distribution ρ flock (red), ρ stream (yellow), and ρ swarm (blue) are shown. The estimation of the black line (data) uses a non-parametric assessment (kernel density estimator) to determine the structure of each zone. AW:0 or AW:1.

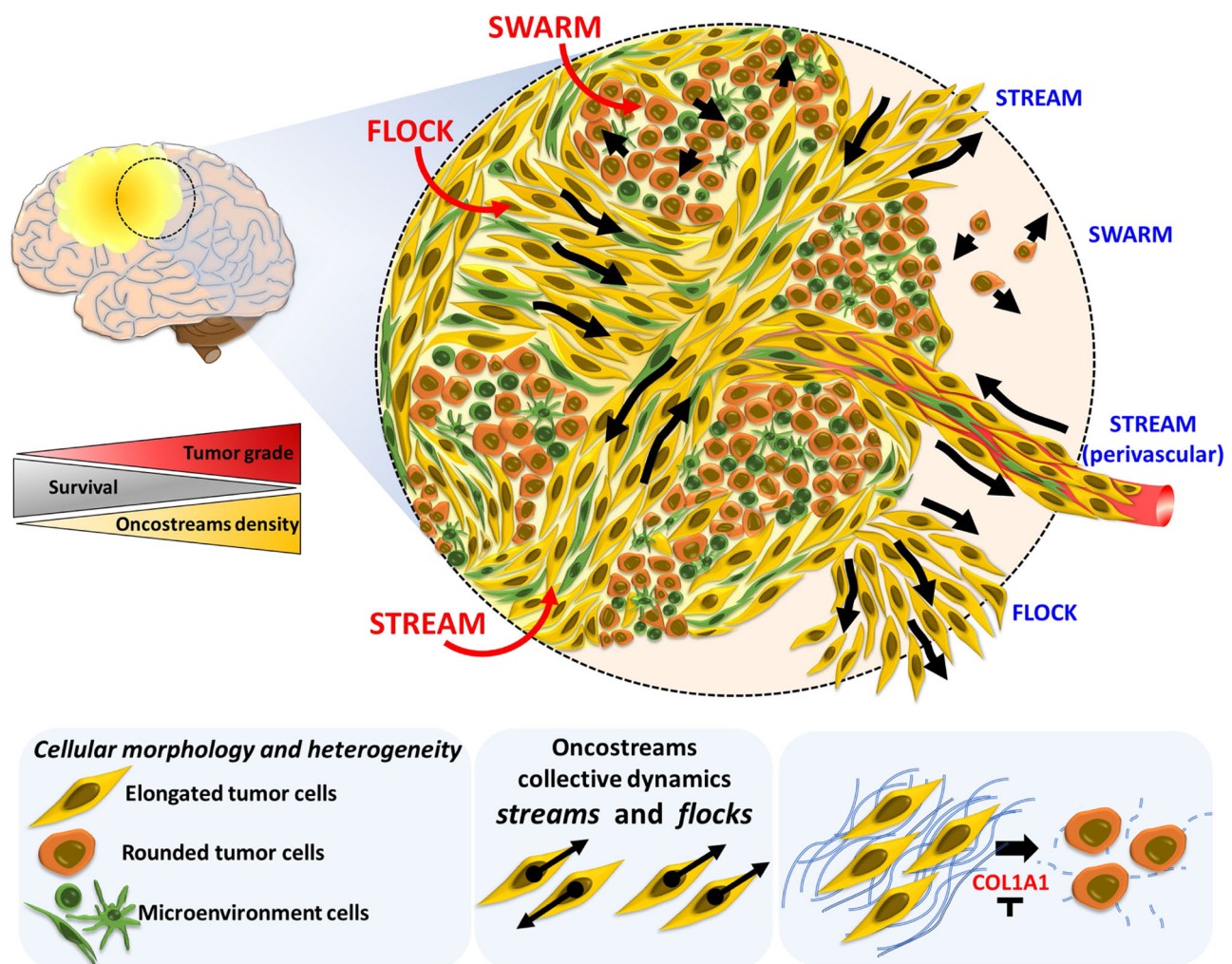

**Fig. 10 Oncostreams are COL1A1-rich multicellular dynamic mesenchymal structures that regulate glioma migration and malignant behavior.** Summary representation of mesenchymal dynamic fascicles (oncostreams) present in high grade gliomas. Our study reveals that oncostreams display directional collective motility patterns including streams and flocks. Non-directional collective motion (swarms) are represented by round cells that move without a preferred direction of motion. Directional dynamic patterns function as tumoral highways to facilitate the intra-tumoral spread of cells and potentially participate in the local invasion of normal brain. Red shows patterns of migration within the tumor core, and blue marks the patterns at the tumor invasive border. Oncostreams are areas of mesenchymal transformation defined by a molecular signature enriched in COL1A1. COL1A1 knockdown disrupts oncostream organization, decreases intratumoral heterogeneity, and significantly increases animal survival. Our study reveals that oncostreams are anatomically and molecularly distinctive, are areas of mesenchymal transformation organized through interactions with the COL1A1 matrix, move by collective motion, and regulate glioma aggressiveness. Figure was created by the authors.

The model was trained using the Adam optimizer with an initial learn rate $a = 0.0001$, $\beta 1 = 0.9$, $\beta 2 = 0.999$, and $\varepsilon = 10E-7$[71]. We used a scheduled learning rate decrease such that the rate was halved every 10 epochs and the model was trained for a total of 75 epochs for each iteration. The best performing model on the mouse dataset was then used as a pre-trained fCNN to initialize training on the TCGA data. We used both mouse and TCGA images for fCNN training with 20% of the TCGA data held out for model validation. Our fCNN was implemented using the model-level Python-based API, Keras (version 2.2.0), with a TensorFlow[74] (version 1.8.0) backend running on two NVIDIA GeForce 1080 Ti graphical processing units. This model is generalizable to both mouse and human images.

**Immunohistochemistry on paraffin embedded brain tumors (IHC-DAB).** Following perfusion with Tyrode saline solution, mouse brains were fixed in 4% paraformaldehyde (PFA), and post-fixed for an additional 48 h at 4 °C. Brains were then processed and embedded in paraffin at the University of Michigan Microscopy & Image Analysis Core Facility using a Leica ASP 300 paraffin tissue processor/Tissue-Tek paraffin tissue embedding station (Leica, Buffalo Grove IL). Tissue was sectioned using a rotary microtome (Leica) set to 5 μm in the z-direction. Endogenous Peroxidase quenching was carried out using a 0.3% $H_2O_2$ incubation for 5 min at room temperature. Heat-induced antigen retrieval was performed using 10 mM citric acid, 0.05% Tween 20, pH 6.0. Tissue permeabilization and blocking were completed using PBS 0.2% Tween-20 with 5% goat serum for one hour at room temperature. Tissue was incubated with primary antibodies at 4 °C overnight at concentrations detailed in Supplementary Table 6. Tissue sections were then incubated with secondary biotinylated antibodies (Polyclonal Goat Anti-Rabbit Immunoglobulins/Biotin, Agilent (E043201-6) at a 1:1000 dilution in PBS with 0.2% Tween-20 overnight at 4 °C. ABC Avidin-Biotin-Complex Binding reagent (Vectastain Elite ABC kit) and Betazoid DAB Chromogen detection kit (BioCare BDB2004) was used according to the manufacturer's instructions.

**Immunofluorescence on paraffin-embedded sections from brain tumors.** Paraformaldehyde fixed and paraffin-embedded tissues were sectioned and then de-paraffinized and re-hydrated. Heat-induced antigen retrieval was performed using 10 mM Citric Acid, 0.05% Tween 20, pH 6.0. Permeabilization was performed using 0.5% TritonX-100 in PBS for 30 min at room temperature while shaking. Tissue sections were blocked in 10% horse serum and 3% BSA in PBS. This method was performed following previous protocol[43]. Primary antibodies were incubated at 4 °C overnight in a humid chamber in 3% BSA in PBS at concertation detailed in Supplementary Table 8. Tissues were then incubated for 90 min at room temperature with Alexa Fluor™ 488, Alexa Fluor™ 555, Alexa Fluor Plus™ 647, or Alexa Fluor 594™ conjugated secondary antibodies (Invitrogen, Thermofisher Scientific) at a dilution of 1:1000 in PBS. Secondary antibodies names and used concentrations are detailed in Supplementary Table 6. COL1A1 detection was performed using HRP-conjugated streptavidin was incubated for 60 min at room temperature. Alexa Fluor™ 488 Tyramide was incubated for 10 min. Nuclei were stained with DAPI (1:1000) in PBS for 5 min.

**Immunohistochemistry on vibratome brain tumor sections.** For immunohistochemistry on vibratome brain tumor sections were left in 4% paraformaldehyde fixation for 48 h and then transferred to PBS 0.1% sodium azide for an additional 24 h at 4 °C. A Leica VT100S vibratome was used to obtain 50 μm coronal brain sections. The immunohistochemistry protocol was performed as described below and as previously performed[75,76]. Vibratome sections were permeabilized in TBS-Triton-X 0.1% for 60 min. Antigen retrieval was performed in 10 mM sodium citrate. Non-specific antibody binding was blocked with 10% goat serum in TBS-Triton-X 0.1% for 1 h at room temperature. Brain sections were then incubated with primary antibody diluted in TBS-Triton-X 0.1%, 1% goat serum, and 0.1% sodium azide for 24 h at RT, in the dark. Sections were then washed 6 times in TBS-Triton-X 0.1% and then incubated with the secondary antibody diluted in 1% goat serum in TBS-Triton-X 0.1% for 24 h at RT, in the dark. Finally, sections were washed 6 times and incubated with 5 μg/ml of 4′,6-diamidino-2-phenylindole (DAPI) (Life technologies, D21490) in PBS for 5 min. Sections were washed again 3 times and mounted on microscope slides with prolong gold anti-fade reagent (Invitrogen, P36930).

**Immunohistochemistry and immunofluorescence imaging.** Images were obtained using bright-field microscopy from independent biological replicates (Olympus BX53 Upright Microscope). Ten different fields of each section were selected at random for study to include heterogeneous tumor areas. For immunofluorescence on paraffin embedded sections from brain tumors images were acquired with a laser scanning confocal microscope (LSM 880, Axio Observer, Zeiss, Germany). Imaged were analyzed using Zen (Blue edition) version 2.5. Integrated density was determined for the analysis of COL1A1 expression using Image J.

**Laser capture microdissection (LCM) of brain tumors.** Malignant glioma tumors were induced by intracranial implantation of dissociated NPA neurospheres in C57BL/6 mice as described above. LCM approach was used to analyze differential mRNA expression of intra-tumoral glioma heterogeneity. This methodology conserves tissue histology integrity and RNA quality[77]. Briefly, animals were perfused with Tyrode solution (0.8% NaCl, 0.0264% $CaCl_2$, 0.005% $NaH_2PO_4$, 0.1% glucose, 0.1% $NaHCO_3$, and 0.02% KCl) for 5 min followed by 30% Sucrose dissolved in Tyrode solution for 15 min. Tissue was maintained overnight at 4 °C in 30% sucrose solution prepared in RNAse free water. Brain tumors were cryopreserved in OCT and stored at −80 °C. Tissue was cryosectioned at 10 μm thickness, placed onto 2 μm polyethylene naphthalate (PEN) and stored at −80 °C. Tumor sections were fixed in ethanol and stained with 4% Cresyl violet and 0.5% eosin dyes dissolved in ethanol. Oncostream areas from three independent mouse glioma samples were dissected from surrounding glioma tissue using a LMD7000 Leica LCM. Tissue was collected in DNase/RNase free 0.5 ml PCR flat head tubes containing 30 μL of RLT lysis buffer and stored at −80 °C until RNA purification.

**RNA isolation, quality control and cDNA library preparation.** RNA was isolated for laser microdissected tissues using the RNeasy Plus Micro Kit following the manufacturer's recommendations (Qiagen). A total tissue area of $2.5 \times 10^6$ to $7 \times 10^6$ μm$^2$ was dissected for each group to increase the efficiency of the mRNA extraction and cDNA library preparation.

0.25 to 10 ng of total RNA was used for cDNA library preparation using a kit suitable for RNA isolation at pico-molar concentrations (MARTer Stranded Total RNA-Seq Kit v2 - Pico Input Mammalian) following manufacturer recommended protocol (Clontech/Takara Bio #635005). Sample volume was adjusted to 350 μL with lysis buffer with 1% β-mercaptoethanol. DNA was removed using gDNA eliminator spin columns. RNA was eluted in 12 μL of RNase-free water warmed at 37 °C.

Before library preparation, RNA was assessed for quality using the TapeStation System (Agilent, Santa Clara, CA) using the manufacturer's recommended protocols. We obtained a RIN between 6 and 7 after laser microdissection of glioma tissue. A RIN of 6 was determined to be suitable for cDNA library preparation.

Final libraries were checked for quality and quantity by bioanalyzer on a Eukaryote total RNA Pico chip using the manufacturer's recommended instructions. The samples were pooled, sequenced on the Illumina HiSeq 4000, as paired-end 50 nt reads with 10% PhiX spike-in, according to the manufacturer's recommended protocols.

**RNA-Sequencing and bioinformatics analysis.** Quality of the raw reads data for each sample using FastQC (version v0.11.3) to identify features of the data that may indicate quality problems (e.g. low quality scores, over-represented sequences, inappropriate GC content).

Tuxedo Suite software package was used for alignment, differential expression analysis, and post-analysis diagnostics by the University of Michigan Bioinformatics Core. Reads were aligned to the reference genome including both mRNAs and Ensemble lncRNAs (UCSC mm10) using TopHat (version 2.0.13) and Bowtie2 (version 2.2.1). Default parameter settings were used for alignment, with the exception of: "b2-very-sensitive" telling the software to spend extra time searching for valid alignments. FastQC was used for a second round of quality control (post-alignment), to ensure that only high quality data would be input to expression quantitation and differential expression analysis. Cufflinks/CuffDiff (version 2.1.1) technique was implemented for differential expression analysis, using UCSC mm10.fa as the reference genome sequence. For expression quantitation, normalization, and differential expression analysis, "—multi-read-correct" was used to adjust expression calculations for reads that map in more than one locus, as well as "compatible-hits-norm" and "upper-quartile–norm" for normalization of expression values. Diagnostic plots were generated using the CummeRbund R package.

Genes and transcripts have been identified as differentially expressed based on three criteria: test status = "OK", FDR ≤ 0.05, and fold change ≥ ± 1.5. Genes and isoforms were annotated with NCBI Entrez GeneIDs and text descriptions. Volcano plots demonstrating total genes and DE genes were produced with the R base package. The Data (significantly impacted GOs biological processes) were analyzed using Advaita Bio's iPathwayGuide[78].

Differentially expressed genes of all tumors were used for gene ontology (GO) analyses and significantly impacted Pathway analyses using Advaita Bio's iPathwayGuide[78]. Network analysis of the DE genes were achieved using Cytoscape and Reactome App. Network was clustered by Reactome Functional Interaction (FI). Analysis of the expression of *COL1A1*, *E-Cadherin*, and *N-Cadherin* in normal tissue and in human gliomas were performed using the dataset of TCGA-GBM and TCGA-LGG from Gliovis (http://gliovis.bioinfo.cnio.es)[47].

**Tumor explant glioma model and time-lapse confocal imaging.** For the analysis of glioma dynamics, we generated tumors by intracranial implantation of $3 \times 10^4$ NPA neurospheres which were used to carry out a 3D explant slice culture glioma model. C57BL6 6–8 weeks old females mice (Taconic Biosciences) were used for the dynamic analyses of the tumor core and B6.129(Cg)-Gt(ROSA)26Sort-m4(ACTB-tdTomato,-EGFP)Luo/J- transgenic 6–8 weeks old females mice (Jackson laboratory, STOCK 007676) were used for the analysis of the tumor invasive border. The red color corresponds to a cell membrane-localized tdTomato (mT) fluorescence expression which is widespread in all cells and tissues. Mice were

euthanized at 19 days' post-implantation for NPA tumors and 31 days' post-implantation for NPAshCOL1A1 tumors. Brains were removed, dissected, and embedded in a 4% agarose solution and kept on ice for 5 min. Then, brains were submerged in ice-cold and oxygenated media (DMEM High-Glucose without phenol red, GibcoTM, USA) and sectioned in a Leica VT100S vibratome (Leica, Buffalo Grove, IL) set to 300 μm in the z-direction. All steps were performed under sterile conditions in a BSL2 laminar flow hood. Brain tumor sections were transferred to laminin-coated Millicel Cell Culture Insert (PICM0RG50, Millipore Sigma, USA) (Supplementary Fig. 23A). Tumor slices were maintained in DMEM F-12 media supplemented with 25% FBS, Penicillin-Streptomycin 10.000 U/mlL at 37 °C with a 5% $CO_2$ atmosphere. After 4–18 h media were replaced with DMEM-F12 media supplemented with B27 2%, N2 1%, Normocin 0.2%, Penicillin-Streptomycin 10.000 U/ml and growth factors EGF and FGF 40 ng/ml. For time-lapse imaging, slices were placed in an incubator chamber of a single photon microscope. We utilized an inverted Zeiss LSM880 laser scanning confocal microscope with AiryScan (Carl Zeiss, Jena, Germany), equipped with an incubation chamber kept at 37 °C with a 5% $CO_2$. To validate the depth of imaging, in some experiments (see Supplementary Fig. 23A–C), before time lapse imaging, we obtained high resolution z-stacks with approximate dimensions of $z = 143.81$, $x = 850.19$, $y = 850.19$. Time-lapse images were then obtained every ten minutes for 100-300 cycles, selecting the imaging plane to fall at the middle of the z-stack to avoid imaging at the bottom of the explant. Following movie acquisition, sections were fixed in 4% paraformaldehyde (PFA) for 2 days. Fixed sections were embedded in 2% agarose for H&E and immunohistochemistry analysis. Sections were processed and embedded in paraffin at the University of Michigan Microscopy & Image Analysis Core Facility using a Leica ASP 300 paraffin tissue processor/Tissue-Tek paraffin tissue embedding station (Leica, Buffalo Grove IL). Tumor explants were used for collagenase treatment. Sections were then treated for 48 h with collagenase (C2399, MilliporeSigma, USA) at a concentration of 5, 10, or 15 units/ml or vehicle control. Following treatment, sections were fixed in 4% paraformaldehyde (PFA) for 2 days.

**Collagenase treatment of tumor explant brain slices model**. Glioblastoma tumors were generated by intracranial implantation of $3.0 \times 10^4$ NPA neurosphere cells into the striatum of C57BL/6 mouse brains. At 19 days following implantation, mice were euthanized per ULAM guidelines. Brains underwent explant sectioning as described above and maintained on Millicel Cell Culture. Sections were then treated for 48 h with collagenase (C2399, Millipore Sigma, USA) at a concentration of 5, 10, or 15 units/ml or vehicle control. Following treatment, sections were fixed in 4% paraformaldehyde (PFA) for 2 days. Fixed sections were embedded in 2% agarose. Sections were then processed and embedded in paraffin at the University of Michigan Microscopy & Image Analysis Core Facility using a Leica ASP 300 paraffin tissue processor/Tissue-Tek paraffin tissue embedding station (Leica, Buffalo Grove IL).

**Cranial window implantation and two photon intravital live imaging in vivo**. Craniotomy and cranial window implantations were performed as described here and in Supplementary Data and following previous protocols described by us and others[23,79,80]. The protocol was conducted according to the guidelines approved by the Institutional Animal Care and Use Committee (IACUC) at the University of Michigan. Briefly, mice were anesthetized and placed in a stereotactic frame. A craniotomy of $3 \times 3$ mm size was made over the right hemisphere between bregma and lambda. $5 \times 10^4$ GFP⁺ NPA glioma cells were intracranially implanted at a depth of 0.8 mm ventral, near the center of the craniotomy overlying the brain cortex of males and females, 6–8 weeks old B6.129(Cg)-Gt(ROSA)26Sort-m4(ACTB-tdTomato-EGFP)Luo/J mice (Jackson Laboratory, STOCK 007676). These mice were utilized to identify (in red) normal brain tissue, and thus establish tumor borders. The cranial window was covered with two round microscope cover glasses, and a metal head bar was positioned on the skull posterior to the cranial window. One week post NPA tumor cells' injection and cranial window implantation, and two-weeks post NPAshCOL1A1 implantation, intravital live imaging was performed using a two-photon microscope (Bruker Technology) with a 20× water immersion objective (Olympus, NA 1.0) for 8–12 h. To avoid imaging at the brain surface, first we acquired high-resolution 3D z-stacks spanning 0–330 μm depth from the surface of the brain. Z-stacks were then imported to Imaris viewer version 9.8 (Bitplane, Imaris, Oxford Instruments, MA, USA) to obtain 3D images. We then used Orthoslicer3D to reveal the depth of the imaging position in relation to the surface of the brain for time-lapse data acquisition. More methodological details are available in Supplementary Methods.

**Mathematical analysis of tumor cell movement**. To determine the movement of cells in different areas of the tumor we performed localized statistical analysis in different zones of the tumor. We selected localized areas based on the organization of cells in clusters, group of cells moving together with similar distribution. Raw data of 4 movies from the tumor core and 4 movies from the tumor border were analyzed for 293 cycles (core) and 186 cycles (border) for a frame rate of $\Delta t = 10$ min between image acquisition. To track cell motion, we used the software Fiji with the plugin TrackMate[81]. Analysis was performed as indicated in detail in Supplementary Methods.

**Classification of glioma migration patterns**. To classify the collective cellular motion behavior of the three types of patterns called flock, stream, and swarm illustrated in Supplementary Fig. 16F we used as criteria the orientation of each cell described by its unique angle velocity denoted θi. More precisely, we transformed the Angle Velocity Distribution graph into a histogram where we examined the distribution of all the values θi. A schematic representation of these distributions is depicted in Fig. 7G. Considering a data-set θn $n = 1…N$ of orientations where $N$ is the total number of cells, θn ϵ [0, 2π] is the direction of the cell n. We tested three types of distributions ρ to describe the dataset and gave a likelihood in each case as described in Supplementary Methods. The Akaike Weight (AW) indicates which pattern has the highest likelihood in each experimental situation[82].

**Statistical Analysis**. All in vivo experiments were performed using independent biological replicates, as indicated in the text and figures for each experiment. Data are shown as the mean ± SEM. Any difference was considered statistically significant when $p < 0.05$. In experiments that included one variable, the one-way ANOVA test was used. In experiments with two independent variables, the two-way ANOVA test was employed. A posterior Tukey's multiple comparisons test was used for mean comparisons. Student's two-sided t-test was used to compare unpaired data from two samples. Survival data were entered into Kaplan–Meier survival curves plots, and statistical analysis was performed using the Mantel log-rank test. Median survival is expressed as MS. Significance was determined if $p < 0.05$. All analyses were conducted using GraphPad Prism (version 8.0.0) or SAS (2021 SAS Institute, Cary, NC). Each statistical test used is indicated within the figure legends.

**Reporting summary**. Further information on research design is available in the Nature Research Reporting Summary linked to this article.

## Data availability

All data associated with this study are in the paper and/or the Supplementary Information. RNA-Seq dataset generated in this study have been deposited at the NCBI's Gene Expression Omnibus (GEO) with identifier GSE188970. Additionally, the following public databases were used: TCGA glioma diagnostic tissue slides from the Genomic Data Commons Portal of the National Cancer Institute [https://portal.gdc.cancer.gov]. Clinical data [http://firebrowse.org]; [http://gliovis.bioinfo.cnio.es]; and [https://portal.gdc.cancer.gov]. TCGA dataset related to Col1A1, E-Cadherin, and N-Cadherin expression and its correlation with patient survival [http://gliovis.bioinfo.cnio.es]. To select shRNA targeting Col1a1 gene, we used the RNAi codex database [http://cancan.cshl.edu/cgi-bin/Codex/Codex.cgi] and InvivoGen's siRNA Wizard [http://www.invivogen.com/sirnawizard]. All the movies/imaging data generated in this study have been provided in the supplementary information. The remaining data are available within the article, supplementary information, or source data file. Cells, plasmids, and other reagents developed in this study could be made available upon request to pedrol@umich.edu. Source data are provided with this paper.

## Code availability

The analysis of oncostreams in mouse and human glioma tissue was performed using U-Net architecture to provide semantic segmentation of specimens using deep learning. Public GitHub repository for the project code can be found at https://github.com/MLNeurosurg/DeepStreams. Analysis of glioma cells dynamics was performed using the Julia Programing Language. Link for this project Script and their dependencies can be found at public GitHub repository https://github.com/smotsch/analysis_glioma.

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

## Acknowledgements

We thank all members of our laboratory, and E.D.Lowenstein, for advice and comments on this work. This work was supported by National Institutes of Health, National Institute of Neurological Disorders and Stroke (NIH/NINDS) grants: R37-NS094804, R01-NS105556, R21-NS107894, R21-NS091555; R01-NS074387 to M.G.C.; National Institute of Neurological Disorders and Stroke (NIH/NINDS) grants: R01-NS076991, R01-NS096756, R01-NS082311, R01-NS122234, R01-NS127378 to P.R.L.; National Institute of Biomedical Imaging and Bioengineering (NIH/NIBI): R01-EB022563; National Cancer Institute (NIH/NCI) U01CA224160; Rogel Cancer Center at The University of Michigan G023089 to M.G.C. Ian's Friends Foundation grant G024230, Leah's Happy Hearts Foundation grant G013908, Pediatric Brain Tumor Foundation grant G023387 and ChadTough Foundation grant G023419 to P.R.L. RNA Biomedicine grant: F046166 to M.G.C. National Cancer Institute (NIH/NCI) grants: R01 CA125577 and R01 CA107469 to CGK. Health and Human Services, National Institutes of Health, UL1 TR002240 to Michigan Institute for Clinical and Health Research (MICHR), Postdoctoral Translational Scholars Program (PTSP), Project F049768 to A.C.

## Author contributions

Conception and design: A.C., M.G.C., and P.R.L. Development of methodology: A.C., M.S.F., P.J.D., A.E.A., T.H., W.N.A.-H., M.L.V., D.B.Z., S.M., and P.R.L. Acquisition of data, analysis, and interpretation: A.C., M.S.F., P.J.D., A.E.A., T.H., M.L.V., D.B.Z., C.A. II, M.G.C., S.M., and P.R.L. Human histopathology analysis and identification of oncostreams: A.C., C.K., A.E.A., and P.R.L. Laser microdissection protocol: A.C., P.R.L., P.E.K., A.K. Development and establishment of intravital imaging using multiphoton microscopy: A.C., M.S.F., P.R.L., G.L.Q., and P.F.A. Development and experimental assistance with human glioma cell lines: C.E.B. Manuscript writing: A.C., S.M., P.R.L. Administrative, technical, or material support (i.e., reporting or organizing data, constructing databases): A.C. and P.R.L. Study supervision: M.G.C. and P.R.L. All authors reviewed the final version of the manuscript.

## Competing interests

The authors declare no competing interests.
