## [Peer review file · Nature Communications]

REVIEWER COMMENTS

Reviewer #1 (Remarks to the Author): Expert in glioma models and genomics

In this manuscript Comba et al., present a novel facet of intratumoral heterogeneity in gliomas. They focus on so-called 'oncostreams', which they describe to represent mesenchymal-like tumor cells involved in the collective invasion. Oncostreams appear particularly high in IDHwt high-grade gliomas (GBMs), and are enriched in mesenchymal tumors. The study was well designed and presented results support the claims. The authors apply histological analyses in glioma patient tumors and experimental models combined with AI-based image analysis. They further assess features of the oncostreams at the transcriptomic and functional levels. They reveal an important role of COL1A1 in supporting the formation of the oncostreams. The presented data and conclusions are novel and will be of interest to the specialists in the field as well as to the wider audience.

Major comments:

1. In the introduction I would suggest to introduce better tumor plasticity and possible reversible transitions between the phenotypic states in high grade gliomas. Although it is well established that mesenchymal-like cells are present in angiogenic and hypoxic niches, the mesenchymal features of invasive cells are less understood. The references supporting this claim are missing in the introduction.
2. The manuscript would profit from the reorganization of the presented results for the wider audience. E.g.:
 - Initial description and analysis of the oncostreams in patient tumors and mouse models is scattered across different figures and is difficult to follow (Fig 1 and 2). Description of the mouse models come after the first results are presented. I would suggest to present the analysis of oncostreams in mouse models and patient tumors on the separate figures respectively.
 - Gene expression analysis of the laser-dissected oncostreams (Fig. 3) and the functional validation of COL1A1 KD (Fig. 6) could be presented together. It is not clear why the authors focus on the NCAM vs. ECAM analysis in the middle of the manuscript (Fig. S15). Was NCAM differentially expressed in the cells forming oncostreams? Is similar cell proliferation supported by the gene expression data?
3. Mesenchymal-like GBM cells are known to interact with tumor-associated macrophages and mesenchymal GBM tumors show higher levels of this microenvironmental component. The authors should present the quantification of macrophages within oncostreams compared to the surrounding areas.
4. The authors present increased survival of mice upon implantation of shCOL1A1 NPA and NPD cells. shCOL1A1 tumors show morphological changes and lack of oncostream features. Where other features of the tumor cells (e.g. tumor cell proliferation, single cell and collective invasion) as well as tumor microenvironment (e.g. structure of blood vessels, presence of tumor associated macrophages) changed? Do shCOL1A1 cells show differential features only in vivo, or is growth/invasion also impaired

in vitro suggesting change in tumor intrinsic properties? These assessments should be presented, to exclude other factors influencing tumor growth in vivo upon COL1A1 KD.

Minor comments:

1. More detailed figure legends regarding choice of markers to stain tumor cells in patient tumors and different experimental models would be beneficial for the general audience (e.g., Fig 1 G-H, Fig. S2). Is GFAP expressed only in tumor microenvironment in the selected mouse models (NPA+NPD or NPD only?) and presented patient tumors? Why is nestin used to detect tumor cells in the NPA model, and GFAP in the NPD model?
2. line 137: I assume that GFP+ tumor cells correspond simply to the genetically engineered mouse tumor cells. Please revise the sentence.
3. line 252-253: Do authors suggest that cell shape is a driving feature or a consequence of the collective motion? Please precise the 'involved' verb here.
4. Fig S22: Please verify the GBM IDHwt subtype analysis. COL1A1 mRNA expression in GBM should be between log₂ 7-20 based on the first graphs, whereas GBM subtype analysis is indicating a gradient between log₂ 4-10.

Reviewer #2 (Remarks to the Author): Expert in mathematical modelling for glioma and image analysis

This manuscript introduces the concept of 'oncostreams' which are observable on histological images and promote the invasion of cancer cells further in the brain. These oncostreams are investigated using biological experiments in conjunction with mathematical modeling and machine learning. Moreover, since oncostreams are consistent with poorer prognosis, the presence of oncostreams could be an important factor in predicting disease progression. When examining genetic markers, these oncostreams were correlated with COL1A1, making COL1A1 a possible biomarker and its depletion a potential therapy. This manuscript is very well written and presents, and defends, a significant and original finding that will be of wide interest to the cancer community. I do believe the manuscript could address a few concerns.

Please add additional details for the deep learning algorithm for the oncostream segmentation. What was the IOU over the testing set? I see that training and validation accuracies are reported in Figure S4, but the testing is not reported. Were two separate models trained to segment each of the datasets? This is what seems likely given the caption for S4, but it is not clear. If so, why not train a more generalizable model that can find oncostreams in both mouse and human images? In Figure S3, the authors state that

a threshold of 0.9 was chosen for binarizing the mask. Was this value optimized as a hyperparameter using the validation set? If so, the authors should state this.

Did the authors investigate whether NPA tumors that did not have oncostreams were any less problematic? For example, in Fig 2D, there are quite a few data points for the NPA tumors that have similarly sized oncostream area % as the NPAI tumors. Was the prognosis different for those tumors that had low oncostreams, even if they were NPA? Similarly, were the high oncostream area% NPAI tumors associated with a worse outcome?

Were the angles and aspect ratios measured for all images or just the select few shown in the figures? This is not clear from the manuscript.

Rewriting the section on glioma migration patterns in the supplementary material for clarity would greatly improve the manuscript and allow for reproduction of results. I will list a few of the sticky points that were difficult to follow in this section/ For the classification of the glioma migration pattern, on line 228-229 it is stated that “The summing in n is necessary to ensure that the distribution is 2π -periodic.” However, the summation is in k . When we look at the likelihood (Eq. 2), n goes from 1 to N (what is N ? Total cells in the image? Total cells in each “zone”? If so, how are the zones determined?) There are some issues with the equation numbering. Equation 2.3 is referenced, but there is no equation 2.3. There is also a self-referencing equation, since on line 243 we are told that is defined in Eq 2.4, which it is not. Maybe there is a missing definition or equation? The example for model selection references Figure 3D (gene expression), when it should reference 4D (cell directions).

Overall, the paper was well written and easy to understand. There were only a few points for specific clarification that I have listed below.

1. The experiments for the human glioma stem cells (MSP-12) was sometimes abbreviated as MSP-12 and sometimes as MSP12. If these are the same cell line, the authors may want to use consistent notation throughout the paper. If they are different, please include a sentence describing their differences.
2. When discussing the knockdown of COL1A1, the authors mention there was increased animal survival. Is this shown in the data (or figures)?
3. Line 176-177: Authors state that “analysis found that oncostreams occupied $15.28 \pm 6.105\%$ of the area in NPA tumors compared with only $1.18 \pm 0.81\%$ in NPAI tumors (Fig. 3C and D, ...)”. When examining the figures, I think these results are shown in Fig 2C and D. Similarly, the authors refer to Fig 3E in the next line, but I think it should be Fig 2E.
4. Line 485: “major” instead of “mayor”

5. Line 488: Please remove the second comma
6. Line 630: Please correct spelling to Akaike (and maybe put a relevant citation for the method).
7. Citation 62 appears to be incomplete.
8. Fig 5G: Some of the subtitles are cut-off.
9. The supplementary tables have page numbers and line markers that the authors may want to remove.

Reviewer #3 (Remarks to the Author): Expert in glioma morphology, dynamics and mouse models

In this manuscript, the authors try to convince the reader that "oncostreams" drive glioma invasion, and try to make a (weak) connection to mesenchymal differentiation and Col1A1. Everything is founded on that assumption.

My problem is that the data provided is not demonstrating this adequately. Similarities between completely artificial tumor cell invasion patterns in/on(?) brain slices (that are notorious for this artificial invasion pattern) are superimposed on human and mouse H&E stained brain sections where morphologically similar structures could be present, or not. Probably not, since these structures look more like biomechanically governed tumor cell "strands" than invasive routes. Functionally, I severely doubt the relevance of this for brain invasion in glioma. In fact, people doing intravital imaging are NOT seeing this invasion pattern, at least not in state-of-the-art animal models of the disease. The authors would need to provide intravital imaging data, or convincing and well-analyzed data from freshly explanted human brain tumors, to convince anyone in the field otherwise / that they have any point here. Therefore, I cannot recommend publication of this study.

Response to reviewers

We thank reviewers for their thoughtful consideration of our manuscript. Below please find an itemized response to each comment (reviewer's comments are in **bold**).

Reviewer #1 (Remarks to the Author): Expert in glioma models and genomics

General comments:

In this manuscript Comba et al., present a novel facet of intratumoral heterogeneity in gliomas. They focus on so-called 'oncostreams', which they describe to represent mesenchymal-like tumor cells involved in the collective invasion. Oncostreams appear particularly high in IDHwt high-grade gliomas (GBMs) and are enriched in mesenchymal tumors. The study was well designed and presented results support the claims. The authors apply histological analyses in glioma patient tumors and experimental models combined with AI-based image analysis. They further assess features of the oncostreams at the transcriptomic and functional levels. They reveal an important role of COL1A1 in supporting the formation of the oncostreams. The presented data and conclusions are novel and will be of interest to the specialists in the field as well as to the wider audience.

We thank the reviewer for their positive comments regarding the design, novelty of the study and the results supporting the role of COL1A1 in the formation of Oncostreams in high grade gliomas.

Major comments:

1. In the introduction I would suggest to introduce better tumor plasticity and possible reversible transitions between the phenotypic states in high grade gliomas. Although it is well established that mesenchymal-like cells are present in angiogenic and hypoxic niches, the mesenchymal features of invasive cells are less understood. The references supporting this claim are missing in the introduction.

As requested by this reviewer we included a new paragraph to better introduce issues related to tumor plasticity and reversible transition between phenotypic states. The new text (page 2, lines 60-67) is in the Introduction section of the revised version of the manuscript, which also includes an expanded list of appropriate references. A reference to the fact that mesenchymal features of invasive cells are less understood has also been included (page 3, lines 73-77).

2. The manuscript would profit from the reorganization of the presented results for the wider audience. E.g.:

- Initial description and analysis of the oncostreams in patient tumors and mouse models is scattered across different figures and is difficult to follow (Fig 1 and 2). Description of the mouse models come after the first results are presented. I would suggest to present the analysis of oncostreams in mouse models and patient tumors on the separate figures respectively.

We truly appreciate this suggestion to help us make our manuscript clearer and more informative for the wider audience of readers. Accordingly, we have reorganized the results section and the corresponding figures of the revised version of the manuscript. The revised manuscript thus includes the following changes:

- To present oncostreams to the reader, Figure 1 A-D introduces the presence and general morphological features, i.e. cell shape and alignment, in mouse and human tissues.
- Figure S1A now introduces the genetically engineered mouse models of glioma.
- Figure 1E-G and Figure 2 then show the main analysis of oncostream biology in mice, which is now been separated from the analysis of oncostreams in human tumors, which is shown in Figure 3. The new text now follows this described outline.

- Gene expression analysis of the laser-dissected oncostreams (Fig. 3) and the functional validation of COL1A1 KD (Fig. 6) could be presented together.

This issue has now been addressed in the revised manuscript. We present together the molecular analysis of oncostreams using laser-capture microdissection (in the **new Fig. 4**); this is immediately followed by the functional analysis of COL1A1 downregulation in GEMM of gliomas (shown in the **new Fig. 5**). The new text now follows the described outline.

- It is not clear why the authors focus on the NCAM vs. ECAM analysis in the middle of the manuscript (Fig. S15). Was NCAM differentially expressed in the cells forming oncostreams?

Mesenchymal transformation is characterized by the overexpression of N-Cadherin. To assess the mesenchymal nature of collective motility in tissues with higher malignancy and density of oncostreams, we analyzed markers for adherens junctions. We did not detect an overexpression of N-cadherin within oncostreams; nevertheless, N-cadherin has been shown to be elevated in high grade gliomas, and correlates with reduced survival. Thus, given the general relevance of N-cadherin to the mesenchymal state, and its potential role in tumor malignancy we would like to keep the reference to N-cadherin expression. These issues are now mentioned in the text in pages 12-13, lines 379-387.

- Is similar cell proliferation supported by the gene expression data?

Yes. We found no differences in cell proliferation within oncostreams compared to non-oncostream areas. This conclusion is also supported by the molecular data of the laser-microdissected oncostreams. No differences were found regarding cell proliferation in the differentially expressed signature of oncostreams. We have now included this information in the Results section of the revised manuscript. Please see page 13, lines 389-392, and **Fig. Supplementary 21D-E, and Fig. 4 A-C**

3. Mesenchymal-like GBM cells are known to interact with tumor-associated macrophages and mesenchymal GBM tumors show higher levels of this microenvironmental component. The authors should present the quantification of macrophages within oncostreams compared to the surrounding areas.

In this study we show that oncostreams are mesenchymal-like heterogeneous multicellular structures that contain both tumoral and non-tumoral cells. As requested by the reviewer, we quantified the density of tumor-associated macrophages (TAM) within oncostreams compared to surrounding areas. Using two different markers for tumor associated microglia/macrophages cells (IBA1 and CD68) we determined that oncostreams exhibit a statistically significantly higher density of TAM compared to outside areas (**Fig. 1F-G**) in GEMM of glioma. In addition, we analyzed the presence of mesenchymal-like cells (ACTA2+), and also observed a significant enrichment of these cells within oncostreams (**Fig. 1E**). We included this information in the Result section pages 5-6, lines 157-166.

4. The authors present increased survival of mice upon implantation of shCOL1A1 NPA and NPD cells. shCOL1A1 tumors show morphological changes and lack of oncostream features.

Were other features of the tumor cells (e.g. tumor cell proliferation, single cell and collective invasion) as well as tumor microenvironment (e.g. structure of blood vessels, presence of tumor associated macrophages) changed? Do shCOL1A1 cells show differential features only in vivo, or is growth/invasion also impaired in vitro suggesting change in tumor intrinsic properties? These assessments should be presented, to exclude other factors influencing tumor growth in vivo upon COL1A1 KD.

In this study we show that COL1A1 downregulation in two GEMM of glioma increase animal survival (**Fig. 5F, G**). We observed that tumors with downregulated COL1A1 exhibit changes in cell morphology (loss of mesenchymal features), and they become depleted of oncostreams. As requested by this reviewer, we now provide further support for the effects of COL1A1 downregulation in glioma biology; specifically, we provide evidence of changes in various features of tumor cells and the tumor microenvironment in the *in vivo* GEMM of glioma and *in vitro* models.

New data are as follows:

- COL1A1 knockdown decreased cell proliferation (PCNA+ cells) (**Fig. 6A-B**). See results page 10, lines 303-304.
- COL1A1 knockdown increased apoptosis through activation of Cleaved-Caspase 3 and inhibition of the anti-apoptotic protein Survivin (**Supplementary Fig. 14A-C**). Please see results page 10, Lines 305-306.
- COL1A1 downregulation reduces the proportion of tumor associated macrophages (CD68+) (**Fig. 6C-D**). See results page 10, lines 307-314.
- COL1A1 downregulation reduces CD31+ endothelial vascular cells (**Fig. 6E-F**). See results page 10, lines 307-314.
- Inhibition of COL1A1 reduced perivascular mesenchymal-like cells (ACTA2+) (**Fig. 6G-H**) page 10, lines 307-314.
- To analyze whether the inhibition of COL1A1 within glioma cells modifies the patterns of invasion we determined the sinuosity of the borders of GFP+ tumors using the Allen-Cahn equation. Please see methodological details in **Supplementary Material and Methods**, Pages 13-14 This analysis revealed an increase in the sinuosity of NPA tumors suggesting that NPA tumors exhibited higher proportion of collective invasion compared to NPA-shCOL1A1 tumors. (**Fig. 8G-H**). Please see Results, page 15, lines 441-446.
- Moreover, we performed time lapse-confocal imaging in brain explant models of implantable NPA-shCOL1A1 tumors. The statistics of the movement disclose that NPA-COL1A1 knockdown tumors display mainly cells moving in isolation, at very low density (**Fig. S29-32**). Angle velocity and the likelihood analysis indicated that the distribution corresponds predominantly to *swarm* patterns (**Supplementary Fig. 29-32 D, E, F**). Please see page 15, lines 447-453.
- As suggested by the reviewer, to evaluate the intrinsic properties of shCOL1A1 tumor cells we assessed *in vitro* cell proliferation and cell migration. We observed that COL1A1-knockdown cells exhibit a significant reduced cell proliferation and reduced cell migration compared to controls (**Fig. S13 A-D**). Please see page 10, lines 299-301

Minor comments:

1. More detailed figure legends regarding choice of markers to stain tumor cells in patient tumors and different experimental models would be beneficial for the general audience (e.g., Fig 1 G-H, Fig. S2). Is GFAP expressed only in tumor microenvironment in the selected mouse models (NPA+NPD or NPD only?) and presented patient tumors? Why is nestin used to detect tumor cells in the NPA model, and GFAP in the NPD model?

We added more detailed information to the Figure Legend as requested regarding IHC markers used in the corresponding experiments. Moreover, we now added new IHC results for GFAP and Nestin markers for both NPA and NPD. None of these markers are exclusive for a specific tumor type (NPA or NPD). See **Supplementary Fig. S2 C-D**.

2. line 137: I assume that GFP+ tumor cells correspond simply to the genetically engineered mouse tumor cells.
This sentence has been revised and corrected to indicate that this is the case; please see page 5, lines 157-158.

**3. line 252-253: Do authors suggest that cell shape is a driving feature or a consequence of the collective motion?
Please precise the 'involved' verb here.**

We have corrected the sentence to now read: "As predicted by our *in silico* model, these results suggest that cell shape, or eccentricity, is a driving feature in the organization of collective motion patterns" (**Fig. Supplementary 16F**), page 12, lines 355-357.

3. Fig S22: Please verify the GBM IDHwt subtype analysis. COL1A1 mRNA expression in GBM should be between log2 7-20 based on the first graphs, whereas GBM subtype analysis is indicating a gradient between log2 4-10.

We have corrected the appropriate graph axis.

Reviewer #2 (Remarks to the Author): Expert in mathematical modelling for glioma and image analysis

This manuscript introduces the concept of ‘oncostreams’ which are observable on histological images and promote the invasion of cancer cells further in the brain. These oncostreams are investigated using biological experiments in conjunction with mathematical modeling and machine learning. Moreover, since oncostreams are consistent with poorer prognosis, the presence of oncostreams could be an important factor in predicting disease progression. When examining genetic markers, these oncostreams were correlated with COL1A1, making COL1A1 a possible biomarker and it’s depletion a potential therapy. This manuscript is very well written and presents, and defends, a significant and original finding that will be of wide interest to the cancer community. I do believe the manuscript could address a few concerns.

We thank the reviewer for their generous comments and providing help towards improving our manuscript during the revision phase. We really appreciated the positive comments including the recognition that “This manuscript is very well written and presents, and defends, a significant and original finding that will be of wide interest to the cancer community”. Please see below for the detailed responses to the queries by **Reviewer #2**.

1. A) Please add additional details for the deep learning algorithm for the oncostream segmentation. What was the IOU over the testing set?

Below are the IOU values over the testing set that were manually annotated. Our model generalized well to both mouse and human data. In addition to having a mean IOU value of 80.3%, our model did not make any false positive errors in TCGA grade II gliomas. We have added this information into the revised version of the manuscript. Please see **Supplementary Material and Methods** section page 5, lines 131-133 and **Supplementary Table S9**.

Dataset	Number of Slides	IOU (% , mean)	IOU (% , sd)	Range (%)
Mouse	30	76.6	14.2	41.1 - 94.4
TCGA II	11	100*	0	100
TCGA III	11	73.6	11.7	54.9 - 87.9
TCGA IV	12	75.5	12.7	52.6 - 97.3
All	64	80.3	11.2	52.9 - 100

New Table S9- * No oncostreams were present in TCGA grade II data and our model did not have any false positive errors for this dataset

1. B) I see that training and validation accuracies are reported in Figure S4, but the testing is not reported. Were two separate models trained to segment each of the datasets? This is what seems likely given the caption for S4, but it is not clear. If so, why not train a more generalizable model that can find oncostreams in both mouse and human images?

This is an excellent observation, and we thank the reviewer for the comment. We began with an annotated mouse dataset for training and validation in order to prove the feasibility of the proposed method for segmenting oncostreams. We subsequently aimed to apply this method to human H&E images using a model that was trained on both mouse and human images, as the reviewer correctly points out. To optimize performance and generalizability, we trained on a small subset of human data in addition to the mouse data to generate a second

model, which was used to generate the results shown in the manuscript. “This model is generalizable to both mouse and human images.” was added to **Suppl. Materials and Methods**, Page 5, lines 141-142.

1. C) In Figure S3, the authors state that a threshold of 0.9 was chosen for binarizing the mask. Was this value optimized as a hyperparameter using the validation set? If so, the authors should state this.

We thank the reviewer for the comment and apologize for not making this clearer in the manuscript. This threshold was tuned during validation as a hyperparameter. We have added this point to the legend of **Supplementary Fig. S3**.

2- Did the authors investigate whether NPA tumors that did not have oncostreams were any less problematic? For example, in Fig 2D, there are quite a few data points for the NPA tumors that have similarly sized oncostream area % as the NPAI tumors. Was the prognosis different for those tumors that had low oncostreams, even if they were NPA? Similarly, were the high oncostream area% NPAI tumors associated with a worse outcome?

We examined this question carefully and were unable to determine a strict correlation between survival of NPA tumors and oncostream % per tumor, within the NPA group. We were able, however, to show correlations between oncostream content and survival if all tumors from all groups were considered together (NPA and NPAI), confirming that oncostream density correlates with tumor malignant behavior. Whether a larger cohort of NPA tumors would allow to highlight an intragroup correlation between oncostream content and survival will be determined in future experiments. We believe oncostreams to be one of a set of glioma biomarkers, such as microvascular proliferation and pseudopalisade necrosis, that can or cannot be present in individual high-grade gliomas.

3- Were the angles and aspect ratios measured for all images or just the select few shown in the figures? This is not clear from the manuscript.

Angles were measured individually for all images used in our manuscript; these data were utilized as a complementary tool to evaluate and identify oncostreams. In particular, the alignment plots and eccentricity values shown correspond to the respective images shown in each figure. We clarified this in the appropriate figure legends.

4- Rewriting the section on glioma migration patterns in the supplementary material for clarity would greatly improve the manuscript and allow for reproduction of results. I will list a few of the sticky points that were difficult to follow in this section/

A- For the classification of the glioma migration pattern, on line 228-229 it is stated that “The summing in n is necessary to ensure that the distribution is 2π -periodic.” However, the summation is in k .

We improved the section on ‘Classification of glioma migration patterns’ (Supplementary Material and Methods, Page 11, lines 304-306). We corrected the sentence indicated by the reviewer to now read as: “Summing in k is necessary to ensure that the distribution ρ^{flock} is 2π periodic in θ ”.

B- When we look at the likelihood (Eq. 2), n goes from 1 to N (what is N ? Total cells in the image? Total cells in each “zone”? If so, how are the zones determined?)

N is the total number of cells in a given zone. “Zones were defined by either varying density, alignment, geographical distribution, and/or movement patterns of cells for each experiment.” This sentence was added to the **Supplementary Materials and Methods**, page 11, lines 293-296.

C- There are some issues with the equation numbering. Equation 2.3 is referenced, but there is no equation 2.3. There is also a self-referencing equation, since on line 243 we are told that is defined in Eq 2.4, which it is not. Maybe there is a missing definition or equation?

As per the reviewer's request, we have corrected the equation numbers in the revised version of the manuscript. Please see **Suppl. Materials and Methods** pages 11-12.

D- The example for model selection references Figure 3D (gene expression) when it should reference 4D (cell directions).

As per reviewer comments we have changed the Figure number of cell direction to its appropriate place in the new Fig. 7D.

Overall, the paper was well written and easy to understand. There were only a few points for specific clarification that I have listed below.

1- The experiments for the human glioma stem cells (MSP-12) was sometimes abbreviated as MSP-12 and sometimes as MSP12. If these are the same cell line, the authors may want to use consistent notation throughout the paper. If they are different, please include a sentence describing their differences.

We thank the reviewer for their comment. We have unified the name of MSP-12 cells all over the manuscript.

2- When discussing the knockdown of COL1A1, the authors mention there was increased animal survival. Is this shown in the data (or figures)?

These results are shown in **Figure 5 F-G** of the revised manuscript.

3. Line 176-177: Authors state that "analysis found that oncostreams occupied $15.28 \pm 6.105\%$ of the area in NPA tumors compared with only $1.18 \pm 0.81\%$ in NPAl tumors (Fig. 3C and D, ...)". When examining the figures, I think these results are shown in Fig 2C and D. Similarly, the authors refer to Fig 3E in the next line, but I think it should be Fig 2E.

Figure numbers have now been corrected in the revised version of the manuscript.

4. Line 485: "major" instead of "mayor". We have corrected the spelling error.

5. Line 488: Please remove the second comma. Comma was removed.

6. Line 630: Please correct spelling to Akaike (and maybe put a relevant citation for the method).

We have corrected the spelling error for Akaike and added a corresponding citation.

7. Citation 62 appears to be incomplete. We have completed the citation for Ronneberger, Olaf et al, 2015. Please check citation 72.

8. Fig 5G: Some of the subtitles are cut-off. Subtitles have now been corrected.

9. The supplementary tables have page numbers and line markers that the authors may want to remove. Line markers have been removed for Supplementary tables pages.

Reviewer #3 (Remarks to the Author): Expert in glioma morphology, dynamics and mouse models

In this manuscript, the authors try to convince the reader that "oncostreams" drive glioma invasion, and try to make a (weak) connection to mesenchymal differentiation and Col1A1. Everything is founded on that assumption.

My problem is that the data provided is not demonstrating this adequately. Similarities between completely artificial tumor cell invasion patterns in/on(?) brain slices (that are notorious for this artificial invasion pattern) are superimposed on human and mouse H&E stained brain sections where morphologically similar structures could be present, or not. Probably not, since these structures look more like biomechanically governed tumor cell "strands" than invasive routes. Functionally, I severely doubt the relevance of this for brain invasion in glioma. In fact, people doing intravital imaging are NOT seeing this invasion pattern, at least not in state-of-the-art animal models of the disease. The authors would need to provide intravital imaging data, or convincing and well-analyzed data from freshly explanted human brain tumors, to convince anyone in the field otherwise / that they have any point here. Therefore, I cannot recommend publication of this study.

We thank the reviewer for taking the time to assess our manuscript. We have addressed the concerns raised as follows:

In our original manuscript we had analyzed tumor cell migration and invasion in explants of brain slices from an implantable model of glioma using time lapse confocal imaging. Please note that in the explant model tumor cells are not added to brain slices, as brain slices already contain implanted growing tumors. At the borders of these tumors, invasion into normal brain was studied.

In addition, as per this reviewer's suggestion we examined the existence of migration patterns of glioma cells by high resolution time lapse intravital imaging using two photon microscopy. NPA glioma cells were intracranially implanted in the brain of tdTomato (mT/mG) mice as described in detail in **Material and Methods** (page 23, lines 702-714) and **Supplementary Material and Methods** section (pages 9-10, lines 243-265). Intravital imaging analysis of glioma cell migration showed that both the tumor core and the invasive borders exhibit organized aligned cells moving collectively with nematic alignment. Angle Velocity and Likelihood statistical analysis showed the existence of collective streams and flocks in using *in vivo* intravital imaging of glioma (**Fig. 9 and Supplementary Fig. 32**), as requested by this reviewer. Please see results in Pages 15-16, lines 460-487 of the corrected manuscript.

Our results are compatible with another recent paper using intravital microscopy in the study of glioma migration (Alieva et al, 2018). This group studied the invasion dynamics in glioblastomas using *in vivo* orthotopic models of glioma xenografts and time-lapse intravital imaging. Their study described two types of invasion patterns at the glioma border, the *invasive margin* of multicellular invading groups of cells, and the *diffuse infiltration* of single cells. We believe that our statistical analyses of three patterns of cell motility (streams, flocks, swarms) are compatible with the cellular configurations shown by Alieva *et al.*, 2018. *Swarms* correspond to diffuse, single cell infiltration, with these cells displaying increased speed and less directionality in both studies. Alieva's *et al.* cells within *invasive margin*, display a pattern akin to what we described as *streams*, as they suggest that cells are moving in two directions, back and forth. As this manuscript did not perform a detailed statistical and quantitative study of cell directionality as described by us, we cannot take the comparisons further. Overall, however, our data are compatible with, and extend, the data of Alieva *et al.*, 2018.

Thus, herein we present a quantitative comprehensive view of the dynamic organization of the tumor border that is compatible with and expands upon the extant evidence of the self-organizing dynamics of the glioma invasive border in *ex vivo* and *in vivo* mouse models of gliomas, including *in vivo* intravital imaging of gliomas.

REVIEWER COMMENTS

Reviewer #1 (Remarks to the Author):

Comba et al. presents now a timely and well organized manuscript that will be of a general interest to the audience of Nature Communications. The new data presented by the authors strengthens the initial claims and explains in more details the function of COL1A1 in shaping oncostreams and GBM heterogeneity in general. The authors have sufficiently addressed my initial comments.

I've noticed a typo in line 224. Please, exchange SOX+ to SOX2+.

Reviewer #2 (Remarks to the Author):

I thank the authors for their revisions and they have adequately addressed all of my concerns.

Reviewer #3 (Remarks to the Author):

The authors have indeed performed additional experiments and include now intravital imaging by 2P microscopy (new Fig. 9).

While this effort is certainly laudable, the data only strengthens my scepticism that the invasion patterns described and analyzed in this entire study ("oncostreams" etc.) are experimental artefacts.

In fact, the data provided looks very much like tumor cells crawling on the very surface of the brain. This reviewer is fully aware that - under these completely artificial conditions - "stream-like" migration patterns do exist, and indeed frequently so. However, this reviewer is also aware that this is NOT AT ALL a reflection of how tumor cells in gliomas move in true / proper brain tissue. Here, none such migration/movement/invasion patterns can be observed when investigating state-of-the-art models.

You can actually tell from (new) Figs. 9G and 9I that the authors image at the very brain surface, in the meningeal region, where no "normal" brain parenchyme cells are present.

To put it into one sentence: the authors provide data of a growth pattern here that is well aware to experts in the field, but that is normally avoided to record at all, because it clearly reflects an artificial growth pattern that is not relevant for the human disease: tumor cells growing in a 2D-environment, sitting on the top of the brain. It is very plausible that this growth pattern also occurs in their other experimental conditions: After brain slicing, the tumor cells tend to overgrow the brain slice with the same artificial growth pattern.

Minor point: The authors should clarify what they believe the "red signal" is showing in 9B and G and I.

Reviewer #4 (Remarks to the Author): Expert in intravital imaging and glioblastoma

GBM rarely grows in the leptomeningeal space, so it is always important to check that the phenomenon shown is not just representative of the very surface of the brain. Even if intravital microscopy is not deep imaging method, it can easily reach 200-500 um. I think the authors need to find a way to clearly demonstrate that the movies shown and quantified are not taken at the very surface of the brain (or on the dura mater). This is always one of the most important controls to be sure that GBM cells are in the proper orthotropic environment (and this fits perfectly with the Rev#3 criticism). Second-harmonic imaging microscopy may be instrumental for example to locate the collagen fibers of the dura to show that the movies shown are not taken at the very surface of the brain.

According to the M&M, the authors do not remove the dura mater during the cranial window setup. This is not a required step, even if performed by several labs working on brain intravital microscopy. Anyway, the dura mater is connective tissue and is highly composed of collagen fibers (almost never present in the brain itself), so it is not surprising that they see the same migration described for COL1A1-rich areas. This conceptually does not disprove everything shown in the manuscript (COL1A1 may be simply produced by tumor cells themselves), but I think that suggests the need of further technical controls.

Thus, overall, I think that the manuscript shows and investigate the interesting phenomenon of GBM oncostreams, but I agree with Rev#3 comments: the intravital microscopy part rises more questions than answers and it needs the right controls to prove that is not just artifactual.

Response to reviewers

We thank reviewers for their thoughtful consideration of our manuscript. Below please find an itemized response to each comment (reviewer's comments are in **bold**).

Reviewer #1 (Remarks to the Author):

Comba et al. presents now a timely and well organized manuscript that will be of a general interest to the audience of Nature Communications. The new data presented by the authors strengthens the initial claims and explains in more details the function of COL1A1 in shaping oncostreams and GBM heterogeneity in general. The authors have sufficiently addressed my initial comments.

We thanks reviewer #1 for the positive feedback about the first revised version of our manuscript, especially regarding the interest of the manuscript for the general audience of Nature Communications and the improvements performed in the new version of the manuscript to explain the role of COL1A1 in the organization of oncostreams patterns in high grade gliomas.

Minor revision: I've noticed a typo in line 224. Please, exchange SOX+ to SOX2+.

This sentence has been corrected in the new version of the manuscript.

Reviewer #2 (Remarks to the Author):

I thank the authors for their revisions, and they have adequately addressed all of my concerns.

We thank the reviewer for the time and effort committed towards improving our manuscript.

Reviewer #3 (Remarks to the Author):

The authors have indeed performed additional experiments and include now intravital imaging by 2P microscopy (new Fig. 9). While this effort is certainly laudable, the data only strengthens my skepticism that the invasion patterns described and analyzed in this entire study ("oncostreams" etc.) are experimental artefacts. In fact, the data provided looks very much like tumor cells crawling on the very surface of the brain. This reviewer is fully aware that - under these completely artificial conditions - "stream-like" migration patterns do exist, and indeed frequently so. However, this reviewer is also aware that this is NOT AT ALL a reflection of how tumor cells in gliomas move in true / proper brain tissue. Here, none such migration/movement/invasion patterns can be observed when investigating state-of-the-art models.

You can actually tell from (new) Figs. 9G and 9I that the authors image at the very brain surface, in the meningeal region, where no "normal" brain parenchymal cells are present.

To put it into one sentence: the authors provide data of a growth pattern here that is well aware to experts in the field, but that is normally avoided to record at all, because it clearly reflects an artificial growth pattern that is not relevant for the human disease: tumor cells growing in a 2D-environment, sitting on the top of the brain. It is very plausible that this growth pattern also occurs in their other experimental conditions: After brain slicing, the tumor cells tend to overgrow the brain slice with the same artificial growth pattern.

We thank the reviewer for taking into consideration the effort of including additional experiments using intravital imaging by two-photon microscopy in the first round of revisions. We believe that addressing the new concerns of this reviewer, have assisted us in strengthening our manuscript

In the revised version of our manuscript, we address the following two main concerns of this reviewer:

- 1. Collective migration patterns of glioma cells are experimental artifacts of moving tumor cells crawling on the very surface of the brain, in the meningeal region, where no "normal" brain parenchyma cells are present (old Fig. 9G and 9I).***
 - 2. After brain slicing, the tumor cells tend to overgrow the brain slice with the same artificial growth pattern.***
1. **A-** Considering the reviewer's concern we replaced all previous intravital movies and we set-up new intravital imaging experiments using two-photon microscopy. As described in Material and Methods, 5×10^5 NPA glioma cells were injected stereotactically in the middle of the craniotomy at a depth of 0.8 mm from the top of the brain surface (New Fig. 9A). Mice were imaged after 7-15 days post-surgery. To confirm that migration analyses was not performed on the top of the brain surface we first acquired high resolution z-stack images of the tumor volume from the top of the surface of the brain until a depth of 300-330 μm (596 μm x 596 μm x 300-330 μm , z-steps of 1 μm). After analyzing the tumor volume, we selected an imaging plane compatible with cellular architecture and image quality necessary to analyze cell migration; we performed time lapse laser scanning two photon imaging at a depth of >100 μm (Please see imaging set up and acquisition procedures in **Fig. 9B-D, Supplementary Fig. S35 and S36**). We performed 3D-Orthogonal reconstructions to show the z-position of the time-lapse plane (For Movie #14: 120 μm [**Fig. 9**], Movie #15: 140 μm [**Supplementary Fig. 35**], Movie #16: 145 μm [**Supplementary Fig. 36**]). The migration analysis of glioma cells growing at these depths shows that cells move following a collective organized migration pattern (i.e., streams), similar to the one previously shown by confocal imaging of brain tumor slices.

B- Furthermore, to show the exact anatomical location of the tumor, following two photon imaging we perfused mice and fixed the brain to perform fluorescence immuno-histochemistry analysis on paraffin embedded 5 μm -thick coronal sections (**Supplementary Fig. S35B-C**). We show in **Supplementary Fig. S35B**, outlined by a white dotted line, the location of the area imaged by intravital microscopy for z-stack acquisition; the time-lapse imaging plane was located at a depth of 140 μm (**Supplementary Fig. S35A**). In this figure (**Supplementary Fig. S35C**) we can identify the surface of the brain (tdTomato, in magenta color) and the location of the tumor (GFP+ cells) growing in the midst of longitudinal parenchymal blood vessels (TdTomato+), and astrocytic processes (GFAP+, in red) (**Supplementary Fig. S35B-C**). Taken together these data show that we are imaging well below the brain surface and within the parenchyma of the brain.

2. To address the concern that in the explant model the tumor cells tend to overgrow the brain slice with an artificial growth pattern we performed the following experiment as shown in **Supplementary Fig. S23**, and described in Material and Methods of the revised manuscript. For this particular experiment, 300 μm -thick tumor slices were maintained in the incubator at 37 °C with a 5% CO₂ atmosphere. After 4 hours, media was replaced and slices were immediately placed in an incubator chamber of a single photon microscope for time-lapse confocal imaging. The microscope utilized is an inverted Zeiss LSM880 laser scanning confocal microscope.

Before starting the time lapse imaging, to validate the depth of imaging within the 300 μm slices, we obtained high resolution z-stacks with approximate dimensions of z=143.81, x=850.19, y=850.19, and z-step intervals of 4.96 μm . After Z-stack acquisition, time-lapse images were obtained every ten minutes for 100-300 cycles, after selecting the imaging plane to fall within the middle of the z-stack to avoid imaging at the bottom of the explant. Reducing the time of the slice incubation and performing 3D z-stack reconstructions we were able to describe the exact same migration patterns found previously in explants and in the intravital models. This demonstrates that the presence of collective motion patterns as '*streams*' are not located at the surface of the explant slices.

Moreover, as shown in **Supplementary Fig. S16**, sections were fixed in 4% paraformaldehyde (PFA) after movie acquisition. We observed that the tissue organization, cellular alignment, and eccentricity seen in histological sections are compatible with the migration patterns found in the time lapse imaging analysis.

Minor point: The authors should clarify what they believe the "red signal" is showing in 9B and G and I.

We incorporated the following sentence to the revised version of the manuscript (see Material and Methods, Line 705). "The red color corresponds to a cell membrane-localized tdTomato (mT) fluorescence expression which is widespread in all cells and tissues". We used transgenic mice B6.129(Cg)-Gt(ROSA)26Sortm4(ACTB-tdTomato,-EGFP)Luo/J (Jackson laboratory, STOCK 007676). By implanting tumors into these animals we can clearly identify the border between GFP+ tumor cells and tdTomato+ brain tissue.

Reviewer #4: (Remarks to the Author):

GBM rarely grows in the leptomeningeal space, so it is always important to check that the phenomenon shown is not just representative of the very surface of the brain. Even if intravital microscopy is not deep imaging method, it can easily reach 200-500 μm . I think the authors need to find a way to clearly demonstrate that the movies shown and quantified are not taken at the very surface of the brain (or on the dura mater). This is always one of the most important controls to be sure that GBM cells are in the proper orthotropic environment (and this fits perfectly with the Rev#3 criticism). Second-harmonic imaging microscopy may be instrumental for example to locate the collagen fibers of the dura to show that the movies shown are not taken at the very surface of the brain.

According to the M&M, the authors do not remove the dura mater during the cranial window setup. This is not a required step, even if performed by several labs working on brain intravital microscopy. Anyway, the dura mater is connective tissue and is highly composed of collagen fibers (almost never present in the brain itself), so it is not surprising that they see the same migration described for COL1A1-rich areas. This conceptually does not disprove everything shown in the manuscript (COL1A1 may be simply produced by tumor cells themselves), but I think that suggests the need of further technical controls. Thus, overall, I think that the manuscript shows and investigate the interesting phenomenon of GBM oncostreams, but I agree with Rev#3 comments: the intravital microscopy part rises more questions than answers and it needs the right controls to prove that is not just artifactual.

We thank this reviewer for suggesting further controls to prove the existence of collective motion patterns in glioma tumors growing in the proper orthotropic environment. These suggestions helped us to make our manuscript clearer and more informative for the wider audience of readers.

In the revised version of our manuscript, we describe new experiments to respond to the following reviewer's concerns:

- 1- *Demonstrate that the movies shown and quantified are not taken at the very surface of the brain (or on the dura mater). Even if intravital microscopy is not deep imaging method, it can easily reach 200-500 μm .***
- 2- *Perform controls to be sure that GBM cells are in the proper orthotropic environment***
- 3- *The dura mater is connective tissue and is highly composed of collagen fibers (almost never present in the brain itself), so it is not surprising that they see the same migration described for COL1A1-rich areas. This conceptually does not disprove everything shown in the manuscript (COL1A1 may be simply produced by tumor cells themselves), but I think that suggests the need of further technical controls.***

1- To prove that the intravital imaging areas and the analyzed migration patterns were not obtained for areas of tumor growing in the leptomeningeal space we performed new intravital imaging experiments at a depth of $>100 \mu\text{m}$ from the top of the brain surface (**Fig. 9, Supplementary Fig. S35 and S36**).

GFP+ glioma cells were implanted at a depth of 0.8 mm in TdTomato transgenic mice. These animals express the cell membrane-localized tdTomato fluorescent protein that is expressed by all cells and tissues in the body. We used these mice to reveal the tumor growing below the brain surface and within the normal brain parenchyma (See details in New Main and Supplementary **Material and Methods** sections).

Taking this reviewer's concerns into consideration and to avoid imaging at the brain surface, we first acquired high-resolution 3D z-stacks spanning 0-330 μm depth from the surface of the brain. Z-stacks were then imported to Imaris viewer version 9.8 to obtain 3D images as shown in **Fig. 9C** and Supplementary Figures

(S35A, S36A, S37A). We then used the Orthoslicer3D function from Imaris to reveal the precise depth of the time lapse imaging plane in relation to the surface of the brain (Fig. 9D, Supplementary Fig. S35A, S36B and S37B).

After 7-15 days of tumor growth, we acquired z-stack images to obtain a 3D orthogonal view of the tumor growing within the cortex. This allowed us to ascertain intravital imaging below the brain surface. We then selected a position at a depth of >100 μm and proceeded to acquire time-lapse images of the tumor growing amidst the normal brain parenchyma at an interval of 5 minutes, for 8-12 hours (Fig 9C-D, Supplementary Fig. S35A, S36A-B and S37A-B). The analysis of cell migration *in vivo* showed that the glioma cells exhibit organized and aligned cells moving collectively at a depth of 120 μm (Movie#14, Fig.9), 140 μm (Movie #15, Suppl. Fig. S35), 145 μm (Movie #16, Suppl. Fig. S36). The depths chosen for each movie depend on the quality of the imaging plane compatible with cellular architecture and image quality necessary to analyze cell movement, i.e. as a tradeoff between image quality and depth of the imaging plane. Imaging depth was always >100 μm . Angle velocity distributions and likelihood analysis determined the presence of collective motion patterns.

2- As a further control and to establish further the exact anatomical location of the tumor growing within the proper orthotopic environment, following intravital imaging, we perfused-fixed the mice, dissected the brains, and performed fluorescence immuno-histochemistry analysis on paraffin-embedded coronal sections; these sections were then imaged by confocal microscopy (Supplementary Fig. S35B-C). The location of the brain surface and the area imaged by intravital microscopy at >100 μm is outlined with white dotted lines (Supplementary Fig. S35B). In this figure, we can identify the location of the tumor (GFP+ cells, green) growing in the proper orthotopic environment containing parenchymal longitudinal blood vessels (TdTomato+, magenta), and astrocytic processes (GFAP+, red) (Supplementary Fig. S35B-C). Thus, together with Supplementary Fig. S35A -which shows the imaging plane at 140 μm -, we demonstrate imaging well below the brain surface and within the parenchyma of the brain (identified as a red fluorescent signal during intravital microscopy [Supplementary Fig. S35A, D]).

3- To prove that collective migration patterns are present in tumors expressing collagen, but are absent in tumors with very low levels of collagen growing in an orthotopic normal parenchymal environment we studied the behavior of NPASHCOL1A1 glioma cells, which only express trace amounts of COL1A1. NPASHCOL1A1 glioma cells were implanted into TdTomato mice at a depth of 0.8 mm. After 15 days, animals were used to analyze tumor migration by intravital imaging. We followed the same procedure as for NPA tumors. First, we obtained high-resolution 3D z-stacks scanning up to 300 μm of depth from the surface of the brain (Supplementary Fig. S37A). We then used the Orthoslicer3D function to reveal the depth of the imaging plane position in relation to the surface of the brain for time-lapse data acquisition. Movie #17 illustrated in Supplementary Fig. S37 was obtained by imaging at a depth of 110 μm (Supplementary Fig. S37 A-B). The intravital migration analysis of NPASHCOL1A1 gliomas showed that glioma cells migrate without a preferred direction and invade the normal brain parenchyma as 'swarms' (Supplementary Fig. S37 C-G). The alignment analysis, Relative Position, and Pairwise Correlation confirm the presence of not-aligned low-density cells compatible with single cell invasion patterns in gliomas with COL1A1 downregulation (Supplementary Fig. S37 E, J, K). Alltogether, these analyses and the previous experiments on NPASHCOL1A1 gliomas using time lapse confocal imaging in the explant model, validate our assertion that not all cells will form organized collective motility patterns, and that expression of COL1A1 by tumor cells determines the formation of high density collective motion patterns.

Further, we showed in previous experiments that depletion of Collagen1A1 from NPA tumor cells eliminates oncostreams and their associated functions. As shown in our manuscript, knockdown of COL1A1 within the tumor cells using genetically engineered mouse glioma models demonstrated that COL1A1 expression is necessary for the formation of oncostream collective patterns. For the benefits of this reviewer, here we provide additional preliminary data of RNAScope *in situ* hybridization of NPA gliomas showing that glioma cells indeed express COL1A1 mRNA within the cells (See below Figure 1).

In summary, in the completely revised and rewritten manuscript we have demonstrated the existence of collective migration patterns (called oncostreams) in both ex-vivo explant imaging within the center of the explant slices, and intravital imaging well below the surface of the brain. Further, we also show that COL1A1 is key to oncostream organization and function. We trust that the reviewers and Editorial Office will agree with our conclusions and extensive controls, and consider this manuscript further for publication.

REVIEWERS' COMMENTS

Reviewer #3 (Remarks to the Author):

The authors have provided additional experiments and analyses with the aim to substantiate their claim that they record in vivo two-photon microscopy from the brain itself and not from the very surface of it.

In the revised Figure 9, they actually now show images that substantiate the existence of oncostreams, but clearly not in a brain tissue environment. This is very solid tumor growing within a lot of pathological = angiogenic vessels. It is almost certain that no "normal" brain cells are in this recorded volume. Of course such a solid, superficially growing tumor mass (which is artificial of course, and frequently happens with this methodology) is not reflecting what is happening in the brain parenchyma itself.

This only further proves my previous main point that "oncostreams", while probably existing in the main = solid tumor mass, are not present (and have nothing to do) with true diffuse brain colonization, i.e. diffuse infiltration of the normal brain with tumor cells which is so typical for the disease.

Since I appreciate the enormous amount of data and experiments that have been performed now, I would suggest the following solution - to allow acceptance of the manuscript, but at the same time to prevent publication of a concept that would be just false:

- the authors need to discuss these aspects in the context of Fig. 9, and the related Video (...if they cannot prove me wrong, by demonstrating neurons, astrocytes, other normal brain structure/cells in the volumes analyzed. However, I am 99.9% certain they cannot do that.)

- the authors need to make crystal-clear, throughout the entire manuscript (title - abstract - results-discussion), that they describe an (interesting) movement pattern of tumor cells within the main tumor mass, but are NOT addressing diffuse brain infiltration/colonization in this manuscript. They need to do that by choosing the right terminology, and also by addressing this point explicitly.

Reviewer #4 (Remarks to the Author):

I was specifically asked to judge the intravital microscopy part of this manuscript.

I raised some concerns on the possibility that the phenomenon shown may be an artifact due to a possible superficial tumor growth.

The authors addressed my points with several controls, multiple new experiments and analyses. I feel now that this part is much stronger and scientifically solid.

This, combined with the rest of the manuscript, makes it a very interesting piece of science. Thus, I definitely recommend it for publication.

REVIEWERS' COMMENTS

Reviewer #3 (Remarks to the Author):

The authors have provided additional experiments and analyses with the aim to substantiate their claim that they record in vivo two-photon microscopy from the brain itself and not from the very surface of it. In the revised Figure 9, they actually now show images that substantiate the existence of oncostreams, but clearly not in a brain tissue environment. This is very solid tumor growing within a lot of pathological = angiogenic vessels. It is almost certain that no "normal" brain cells are in this recorded volume. Of course such a solid, superficially growing tumor mass (which is artificial of course, and frequently happens with this methodology) is not reflecting what is happening in the brain parenchyma itself.

Answer:

We were pleased to see that reviewer #3 agrees with the 'existence of oncostreams'. In addition, this reviewer suggests 'that no "normal" brain cells are in this recorded volume of Figure 9'. In answer to this comment, we agree with the reviewer that Figure 9 represents oncostreams within the main core of the glioma.

Further, in relation to the absence of normal brain cells from our recordings, we agree that using multiphoton microscopy, even when tumors are implanted into td-Tomato mice, it is not possible to identify individual normal brain cells in the area of recording.

Nevertheless, we provide further figures showing that oncostreams within the tumor core are actually embedded within normal brain cells:

-Figure 3E (showing GFAP+ astrocytes within oncostreams in human gliomas);

- Figure S1F (showing positive neurofilament staining, i.e., neuronal axons, within and surrounding oncostreams);

-Figure S2B (showing GFAP positive staining within oncostreams), and S2D (showing oncostreams growing within the context of normal GFAP cells).

-Figure S35D (showing green tumor cells intermingling with normal brain –shown in red fluorescence-).

This only further proofs (sic) my previous main point that "oncostreams", while probably existing in the main = solid tumor mass, are not present (and have nothing to do) with true diffuse brain colonization, i.e. diffuse infiltration of the normal brain with tumor cells which is so typical for the disease.

Answer:

We agree with the reviewer that "oncostreams" exist in the main tumor mass, and are not related to diffuse infiltration (e.g. single cell infiltration) of the normal brain. Nevertheless, we do find oncostreams at tumor borders, i.e., areas of multicellular fascicles of tumor cells at the interphase between tumor and normal brain, as exemplified by the data shown in:

-Figure 8A-C (movie #5), showing oncostream collective migration patterns at the border between tumor cells (green), and normal brain tissue (red).

-Figure 8G, showing oncostreams are shown at the border in NPA tumors but are lost in NPAsCol1A1, while single cell invasion corresponding to diffuse infiltration is seen in both conditions;

-Figure S25A (movie #8), S26 (movie #9), showing oncostreams present at the tumor border;
-Figure S35D (movie #15), showing oncostreams at the border between the tumor cells (green), and normal brain tissue (red).

Similar structures and associated movement patterns were also defined in Alieva et al., especially in their Figure 1; Alieva et al., *Intravital imaging of glioma morphology reveals distinctive cellular dynamics and contribution to tumor cell invasion. Scientific Reports, 9:2054 (2019)* (<https://www.nature.com/articles/s41598-019-38625-4>)

Since I appreciate the enormous amount of data and experiments that have been performed now, I would suggest the following solution - to allow acceptance of the manuscript, but at the same time to prevent publication of a concept that would be just false:

- the authors need to discuss these aspects in the context of Fig. 9, and the related Video (...if they cannot prove me wrong, by demonstrating neurons, astrocytes, other normal brain structure/cells in the volumes analyzed. However, I am 99.9% certain they cannot do that.)

- the authors need to make crystal-clear, throughout the entire manuscript (title - abstract - results- discussion), that they describe an (interesting) movement pattern of tumor cells within the main tumor mass, but are NOT addressing diffuse brain infiltration/colonization in this manuscript. They need to do that by choosing the right terminology, and also by addressing this point explicitly.

Answer:

We accept these comments by reviewer #3. Specifically, we now “make crystal clear” that we “describe an (interesting) movement pattern of tumor cells within the main tumor mass, but are NOT addressing diffuse brain infiltration/colonization in this manuscript.” We clarified throughout the manuscript that we describe the existence of multicellular oncostreams that are present within the main tumor mass, and that they are also present at tumor invasive borders. Yet, oncostreams do not participate in diffuse brain infiltration/colonization. They might play a role in collective multicellular invasion. This difference between diffuse brain infiltration/colonization, and collective multicellular invasion, has now been clarified throughout the manuscript.

Reviewer #4 (Remarks to the Author):

I was specifically asked to judge the intravital microscopy part of this manuscript. I raised some concerns on the possibility that the phenomenon shown may be an artifact due to a possible superficial tumor growth. The authors addressed my points with several controls, multiple new experiments and analyses. I feel now that this part is much stronger and scientifically solid. This, combined with the rest of the manuscript, makes it a very interesting piece of science. Thus, I definitely recommend it for publication.

Answer:

We thank this reviewer for their positive comments.